# How does the institutional environment improve the entrepreneurial quality of returnees? A configuration analysis based on a complex system view

Yiyang Shen[1], Hongtao Yang[1]*, Qiuhua Zhu[2]

**1** School of Business Administration, Huaqiao University, Quanzhou, Fujian, China, **2** Business School, Quanzhou Normal University, Quanzhou, Fujian, China

* yht@hqu.edu.cn

## Abstract

Enhancing high-quality entrepreneurship among returnees stands as a pivotal mechanism in driving the robust economic development of emerging economies. The inquiry into crafting an enabling institutional framework to bolster the excellence of returnee entrepreneurship has garnered significant interest within academia and pertinent industries. Existing research primarily explores the impact of single institutional elements on returnee entrepreneurship, overlooking the diverse pathways to improve its quality under institutional complexity. Building on these insights, the current study explores the complex relationship between institutional configurations and the quality of returnee entrepreneurship in China. It also employs Necessary Condition Analysis (NCA) and dynamic Qualitative Comparative Analysis (QCA) methods based on a complex systems perspective. The research samples include 30 provinces in China. The findings reveal that while individual institutional factors do not, on their own, drive high-quality entrepreneurship among returnees in Periods 1 and 2, the importance of market environment and digital infrastructure becomes pronounced by Period 3. In the initial period, resource-driven entrepreneurship paired with emerging opportunities fosters high-quality outcomes. By Period 2, two additional configurations take shape: legitimacy-driven entrepreneurship under opportunity emergence and a dual-driver model that integrates both opportunities and resources. This dual-driver approach remains the dominant pathway for returnees in Period 3. Additionally, the market environment remains a critical factor across all periods. Digital infrastructure has become increasingly crucial for returnee entrepreneurship. Initially, the market environment focused primarily on financial services, but its connection with digital infrastructure intensified over time. In that regard, differences in economic resources and development across provinces have also led to region-specific pathways for improving the quality of returnee entrepreneurship. The findings contribute to a nuanced comprehension of the disparate progress of returnee entrepreneurial

**Data availability statement:** All relevant data are within the paper and its Supporting Information files.

**Funding:** National Social Science Foundation of China (19BSH110); Huaqiao University's Academic Project Supported by the Fundamental Research Funds for the Central Universities (HQHRZX-202202)

**Competing interests:** The authors have declared that no competing interests exist.

endeavors across regions within emerging economies. Most importantly, they offer theoretical insights to enhance the institutional framework in diverse regions, fostering the attainment of high-quality returnee entrepreneurship.

## 1. Introduction

Returnee entrepreneurs hold significant potential to enrich emerging markets through the introduction of advanced technologies, substantial capital investments, and cross-cultural insights [1,2]. As key contributors to economic growth in these regions, their ventures increasingly benefit from evolving policy frameworks that incentivize domestic entrepreneurship [3,4]. However, companies founded by returnees often encounter challenges stemming from their external status and limited contextual relevance. These obstacles can slow their development compared to domestic companies, impeding their ability to achieve high-quality entrepreneurship [5]. The role of the institutional environment is crucial for entrepreneurial development and economic growth [6,7], provided that high-quality entrepreneurial activities are the core engine of high-quality economic development [8]. Notably, there is a significant geographic concentration of high-quality returnee entrepreneurial activities in China, as evidenced by the "2021 China Overseas Returnees' Innovation and Entrepreneurship Ranking", where 70% of the top 20 companies are located in first-tier cities. This underscores the urgency of optimizing the institutional environment to narrow regional development disparities and enhance the quality of returnee entrepreneurship. Therefore, the research question of this paper is: How does the institutional environment in emerging economies promote the entrepreneurial quality of returnees?

Existing scholarship predominantly examines organizational-level determinants such as psychological resilience and social capital [5,9,10], while underemphasizing macro-institutional architecture. Although some studies acknowledge returnee ventures' embeddedness in formal and informal institutions [11], with particular attention to policy effectiveness and governance quality [12]. While current research has made good progress on the topic, there is still room for further expansion.

First of all, Current research on institutional influences in entrepreneurship remains constrained by disciplinary silos, predominantly relying on sociological institutional theory [13]. While existing studies effectively elucidate how institutional legitimacy and resource allocation mechanisms shape entrepreneurial outcomes, two critical theoretical limitations persist. On the one hand, the exclusive focus on normative-cognitive institutional forces overlooks transaction costs in entrepreneurial decision-making. On the other hand, the lack of cross-paradigm dialogue between sociological and economic institutional theories hinders a comprehensive understanding of institutional complexity [13,16]. This study innovatively combines institutional theories from both sociology and economic theory. It is grounded in sociological institutional theory, highlighting key drivers of high-quality returnee entrepreneurship, including organizational legitimacy and the alleviation of institutional barriers

[10,14,15]. It also integrates the economic transaction cost theory to show how the institutional environment improves entrepreneurial quality. By reducing information asymmetry and lowering transaction costs, it facilitates the flow of technological innovation resources [16]. This integrated perspective provides a more nuanced analytical framework for explaining variations in entrepreneurial activities across regions.

Furthermore, atomized examinations of institutional elements [17,18] fail to capture their synergistic effects on entrepreneurial quality—a gap addressed through our complex systems lens, which reveals co-evolutionary patterns among legitimacy, barriers, resources, and transaction costs [19]. At the same time, existing research focuses more on the impact of informal institutions on returnee entrepreneurship. This results in neglecting the complementary or substitutive effects that may exist between formal institutions and their support structures and informal institutions [20,21]. In different entrepreneurial contexts and levels of economic development, the interaction between formal and informal institutions can vary significantly. This heterogeneity influences regional entrepreneurial activities. This paper incorporates formal and informal institutions into the same institutional framework to analyze their interactive effects.

Finally, existing research employing the dynamic Qualitative Comparative Analysis (Hereinafter referred to as dynamic QCA) method tends to focus on static relationships. This approach often overlooks the dynamic impact of the temporal dimension on outcomes [4]. This paper examines the dynamic relationships between antecedents and outcomes by introducing the time factor. It also fills the existing research gap and provides an empirical foundation and theoretical support for understanding the antecedent dynamics in the process of improving the quality of returnee entrepreneurship.

Building on these insights, the current study adopts a "configurational perspective" focusing on the quality of returnee entrepreneurship across 30 provinces and municipalities in China. It employs Necessary Condition Analysis (Hereinafter referred to as NCA) and dynamic QCA to create a research framework that examines the relationship between the institutional environment and the quality of returnee entrepreneurship. The study identifies the driving pathways for high-quality returnee entrepreneurship across different periods. Additionally, it clarifies the distinct configurational conditions and mechanisms influencing the varying impacts on the quality of returnee entrepreneurship among provinces and municipalities over time.

## 2. Literature review and research framework

Institutions, as human – devised constraints, shape political, economic, and social interactions, thus defining entrepreneurial boundaries. North [20] categorizes the institutional environment into formal and informal types. Formal institutions, per Webb et al. [22], cover laws, regulations and supportive elements like regulatory bodies, financial capital, and infrastructure. Informal ones encompass societal norms, values, and beliefs, determining social acceptability [20]. Institutions' heterogeneity leads to diverse impacts on entrepreneurship across contexts, with some environments being influential and others not [23,24].

Entrepreneurs returning from abroad face challenges distinct from those encountered by domestic startups, embedding them within a unique institutional environment [11,25]. Firstly, compared to domestic companies, returnee entrepreneurial firms have an 'outsider disadvantage', as they lack insight into the domestic market and policies, thus making it difficult to identify high-quality entrepreneurial opportunities and obtain organizational legitimacy and entrepreneurial resources [17,26]. As the gap between Chinese economic development and developed countries narrows, the recognition attitude of the Chinese people toward returnee entrepreneurs has also changed, exacerbating the negative impact of "this outsider disadvantage" [27]. Only with the support of government agencies, social culture, and capital can returnee entrepreneurial firms accelerate learning domestic knowledge, embed into domestic networks, and overcome the outsider disadvantage [28,29]. Secondly, compared to domestic companies, returnee entrepreneurial firms have the disadvantage of "low situational relevance". Although returnee entrepreneurs generally possess high levels of human capital and advanced technology, the inconsistency between domestic and international technology application scenarios makes it difficult for their advanced overseas technology to integrate into the domestic market [11]. A favorable market environment and

advanced digital infrastructure can help returnee entrepreneurs interact closely and more frequently with other members, thereby reducing the obstacles brought by situational inconsistency. It is noteworthy that during the COVID-19 pandemic, the demand for digital infrastructure among overseas returnee entrepreneurial companies surged to a great extent. During this period, the role of digital infrastructure became increasingly prominent. It not only effectively reduced the various obstacles caused by contextual inconsistencies but also substantially decreased the various cost expenditures in the process of entrepreneurship [30]. This change fully demonstrates that in response to global crises, digital infrastructure is important in supporting and promoting the development of overseas returnee entrepreneurial companies. In addition, the disadvantage of 'low situational relevance' also increases the risk of technology knowledge property leakage for returnee entrepreneurial firms, making legal disputes more imminent. A sound legal policy environment can provide legal protection for the patented technology of returnee entrepreneurial firms, increasing the growth dividends brought by technological innovation. In summary, this paper focuses on the unique context of returnee entrepreneurship, while comprehensively considering institutional environmental factors such as government scale, market environment, financial capital, legal policy, social culture, and digital infrastructure. Additionally, it studies the complex relationship of returnee entrepreneurship quality and analyzes the mechanisms and pathways by which the institutional environment configuration comprehensively affects the quality of returnee entrepreneurship.

## 2.1. Methodology and mechanism for achieving entrepreneurial excellence among overseas returnees

Drawing from institutional theory and transaction cost theory, prior research has identified that enhancing organizational legitimacy, allocating entrepreneurial resources and reducing institutional barriers as three pathways to improve the quality of returnee entrepreneurship [10,14,15,31]. However, these studies have analyzed the isomorphic effects of the institutional environment on entrepreneurial activities solely from the sociological perspective of institutional theory, neglecting the transaction cost issues that institutions impose on entrepreneurial activities from an economic perspective [32]. Gehman et al. [13] argue that existing research lacks dialogue between different research perspectives, making it difficult to systematically explain how institutions affect entrepreneurship. Based on this, the present paper integrates the approaches of sociology and economics on the topic within institutional theory to analyze four driving mechanisms that impact the quality of returnee entrepreneurship and examines their synergistic effects through a complex systems lens. These theories provide theoretical support for the subsequent study of how institutional environmental elements affect the quality of returnee entrepreneurship through complex interaction mechanisms.

### 2.1.1. Institutional theory.
The institutional theory offers a non-economic explanation for organizational behavior and strategy. It emphasized that organizational structure and behavior are largely determined and legitimized by the surrounding environment [33]. Institutions serve as the foundation for economic activities by establishing the rules of the game that regulate firms' production, exchange, and distribution activities. Within this framework, firms must adhere to established rules and systems to obtain legitimacy and mobilize their social, economic, and political resources. This way, they can adapt to specific institutional environments and enhance entrepreneurial performance. For returnee entrepreneurs, the institutional environment is a critical factor that cannot be ignored in the entrepreneurial process. The institutional environment fundamentally shapes the contextual framework for returnee entrepreneurship, serving as both an enabler and a constraint on entrepreneurial activities. This environment encompasses formal regulatory structures, such as laws, regulations, and policy systems, as well as informal social norms and cultural practices [34].

Institutional theory suggests organizational legitimacy is crucial for new ventures' survival and growth, directly impacting returnee entrepreneurial firms [35]. Influenced by international environments, returnee entrepreneurs face re – embedding barriers in institutions, cognition, and culture, hindering startup legitimacy [10,11]. The institutional environment, via legal policies and social norms, defines legitimate behavior, lowers legitimacy thresholds, helping returnee firms overcome resource-acquisition hurdles, and boost entrepreneurship quality [28]. A favorable environment reduces legitimacy thresholds, enhances entrepreneurship quality through supportive legal policies and inclusive culture.

 

Institutional theory indicates resource allocation between productive and unproductive entrepreneurial activities affects productive endeavor quality [36], like high-quality returnee entrepreneurship. Returnee firms face "outsider disadvantages" and "low context relevance", impeding domestic network integration and access to essential entrepreneurial resources [5]. A well-structured institutional environment can efficiently allocate resources (finance, fiscal support, infrastructure), channeling entrepreneurial spirit into productive areas, and driving high-quality returnee entrepreneurship [10,37].

Institutional theory posits reducing institutional barriers to entrepreneurial activities can create many high-quality entrepreneurship opportunities for returnees. These barriers include entry, growth, and exit barriers [38,39]. Firstly, returnee entrepreneurs encounter higher entry barriers. A favorable sociocultural environment can weaken undue favoritism toward established and domestic firms, creating opportunities for returnee talents [15]. Secondly, returnee entrepreneurs, with high human capital, often pursue growth via innovation like technological innovation [9]. A fair competitive environment and solid legal frameworks can reduce large corporations' advantages, enhancing highly skilled returnees' high-quality entrepreneurship prospects. Finally, returnee entrepreneurship is mostly driven by a desire to contribute to their homeland or fulfill personal values, heightening their fear of failure. Lowering exit barriers can somewhat ease the loss of individual legitimacy, encouraging them to engage in high – innovation, high-risk, high – quality entrepreneurship [39]. Thus, a well-structured institutional environment should lower institutional barriers, improving returnee entrepreneurship quality through a fairer market and more inclusive sociocultural context.

In summary, the institutional environment significantly impacts returnee entrepreneurship quality. Three key pathways for improvement are enhancing organizational legitimacy, allocating entrepreneurial resources, and reducing institutional barriers. Returnee entrepreneurs must understand trends in domestic and international institutional environments, flexibly adjust strategies and resource allocations, and tackle complex, changing institutional challenges. In short, by continuously boosting adaptability and innovation, returnee entrepreneurs can stand out in fierce market competition and achieve success.

**2.1.2. Transaction cost theory.** The transaction cost theory, created by Coase and developed by Williamson, points out that in addition to the prices of goods, market transactions also involve costs related to information search, negotiation, contracting, execution, and monitoring [40]. These costs affect the operational efficiency and market competitiveness of companies. However, an effective institutional design can reduce transaction costs and promote efficient market operations and enterprise development. Due to discrepancies in educational and work backgrounds, returnee entrepreneurs often face information asymmetry with domestic markets, policies, and social networks [41]. This increases the difficulty of entrepreneurship and the risk of opportunism, leading to higher transaction costs [16]. A favorable institutional environment is crucial for reducing transaction costs for returnee entrepreneurs [42]. In that regard, an open set of transparent government policies and a good environment of social trust can reduce information asymmetry and support returnee entrepreneurs in lowering their transaction costs. A good legal and policy environment can provide adequate legal protection for returnee entrepreneurs, reduce disputes and litigation costs, and stimulate innovation. Additionally, government entrepreneurship support policies can reduce the financial costs for returnee entrepreneurs through funding subsidies, tax incentives, and other methods.

In summary, the institutional environment can effectively reduce transaction costs for returnee entrepreneurship, stimulate innovation, and enhance the quality of entrepreneurship. This can be accomplished by reducing of information asymmetry, strengthening legal protection, and providing financial support.

**2.1.3. Complex system views and returnee entrepreneurial qualities.** The complex systems perspective highlights that market entities are interconnected, and interactive, and economic activities are ever-changing and intricate. In this context, it seems necessary to seek multiple solutions rather than an optimal equilibrium in addressing complex management issues [19]. Therefore, it is essential to identify diverse driving pathways to enhance the quality of returnee entrepreneurship. Moreover, the complex systems view suggests that inductive reasoning is an effective way to analyze complex problems, necessitating the use of new methodologies for in-depth exploration [43].

From the sociological perspective of institutional theory and the economic perspective of transaction cost theory, this paper discusses four driving mechanisms of returnee entrepreneurship quality, providing explanatory mechanisms for analyzing the impact of institutional environmental elements on the quality of returnee entrepreneurship. However, according to the complex systems view, different regions may evolve distinct institutional environmental ecologies, promoting entrepreneurship quality through various combinations of elements. Yet, there is currently a lack of research from a complexity perspective on the intricate relationship between institutional environmental configurations and entrepreneurship quality.

### 2.2. Institutional environment and returnee entrepreneurial quality

Existing research focuses on analyzing the relationship between individual elements of the institutional environment and returnee entrepreneurship. This paper, based on a review of previous studies, provides a theoretical basis for defining the institutional environmental elements that affect the quality of returnee entrepreneurship and for analyzing the relationship between the two.

**2.2.1. Government scale and the returnee entrepreneurial quality.** Government size reflects the extent of support, such as funding and other resources, that domestic authorities provide to returnee entrepreneurial firms [44]. On one hand, the scale of government affects the transaction costs of returnee entrepreneurial firms. Government support through entrepreneurial loans and rent reductions aids returnee entrepreneurs in enhancing their innovation inputs and outputs, thereby improving the quality of entrepreneurship [28]. However, some scholars argue that a larger government size, implying higher public expenditure, may lead to higher taxes for entrepreneurs, increasing the transaction costs for businesses and hindering the enhancement of entrepreneurial activity quality [45]. On the other hand, government size affects the institutional barriers faced by returnee entrepreneurial firms. Expansionary government spending policies can reduce entry barriers for returnee entrepreneurial firms by improving the level of regional public services, thus promoting entrepreneurial activities. Simultaneously, the size of government might suppress the entrepreneurial spirit of returnees [46], casting doubt on the effectiveness of government size in improving entrepreneurship quality.

**2.2.2. Market environment and returnee entrepreneurial quality.** The market environment, acting as an 'invisible hand', significantly influences entrepreneurial activities. A larger market size can lower market entry barriers, reducing the perceived uncertainty among returnee entrepreneurs, and thereby facilitating their engagement in high-quality entrepreneurship [47]. Nevertheless, a favorable market environment can decrease the market transaction costs for returnee entrepreneurial firms, encouraging them to engage in activities such as technological innovation, thus, enhancing the quality of entrepreneurship [48]. However, some scholars argue that a larger market size, reflecting a greater demand for labor, may increase the human capital costs associated with returnee entrepreneurship [49]. In this context, the market environment may have a double-edged sword effect on the quality of returnee entrepreneurship.

**2.2.3. Financial capital and returnee entrepreneurial quality.** Financial capitals indicate the degree to which returnee entrepreneurs can access financial resources, such as bank loans and venture capital, crucial for entrepreneurial endeavors [50]. Due to a lack of collateral, legitimacy, and the dilemma of information asymmetry, returnee entrepreneurs often struggle to obtain external financial resources. This scarcity of financial resources may hinder high-quality entrepreneurship from securing investments that support innovation and growth activities [51]. Therefore, robust financial capital can facilitate the allocation of financial resources, by promoting the development of more productive, high-quality activities. Additionally, the reverse cultural shock experienced upon returning can adversely affect the mindset of returnee entrepreneurs [14]. The greater the financial support for entrepreneurial activities from financial capitals, the higher the perception of environmental safety and stability among returnee entrepreneurs [52], which helps lower institutional entry barriers and motivates them to undertake higher risks in pursuing high-quality entrepreneurial ventures.

**2.2.4. Legal policy and returnee entrepreneurial quality.** Legal policies reflect the degree of protection provided by regional intellectual property rights policies for entrepreneurial activities. First of all, a positive and promising legal and

policy framework can significantly reduce the risk coefficient for returnee entrepreneurs in the entrepreneurial process. This can be achieved by weakening institutional barriers [18,53]. This reduction in barriers broadens the horizons of returnee entrepreneurs, enabling them to access more high-quality entrepreneurial opportunities. It also further motivates them to engage in high-quality entrepreneurial practices [15]. Furthermore, legal policies play a crucial role in the allocation of entrepreneurial resources. Some industrial policies designed to support existing companies may concentrate resources in mature firms, creating obstacles for newly established returnee startups [15]. Additionally, a robust legal system can inadvertently direct resources toward lower-quality entrepreneurial projects [54]. This indicates that the precision and orientation of legal policies are important in resource allocation. Finally, legal policies appear as important in directly influencing the level of transaction costs by ensuring clarity and transparency in property rights [55]. In such a context, the risks faced by new entrants due to bounded rationality and limited reliability are effectively reduced, achieving significant savings in transaction costs [16,50,56]. For returnee entrepreneurs, this means they can more efficiently and securely utilize resources to promote the steady development of their entrepreneurial projects.

**2.2.5. Sociocultural environment and returnee entrepreneurial quality.** A sociocultural environment mirrors the degree of trust and inclusivity within a community. The main driving force behind returnee entrepreneurship is their possession of advanced technologies and business models. However, cultural disparities pose challenges to the integration of their knowledge and technology into the domestic environment, leading to legitimacy deficits and formidable entry barriers [11]. An inclusive sociocultural environment can enhance the cognitive legitimacy of returnee entrepreneurial firms and reduce institutional entry barriers, promoting the development of high-quality entrepreneurial activities [15,57]. Additionally, in areas with high levels of social trust, returnee entrepreneurs are more likely to obtain resources and psychological support from social networks [29]. This can reduce the exit barriers and transaction costs of entrepreneurial activities to some extent, thereby improving the quality of entrepreneurship [26].

**2.2.6. Digital infrastructure and returnee entrepreneurial quality.** Digital infrastructure, as the cornerstone of the development of the digital economy, is contributing to high-quality development, taking over from transportation infrastructure. On one hand, digital infrastructure accelerates the interconnectivity of information and resources within a region, reducing the transaction costs for returnee entrepreneurial firms due to information asymmetry [58]. On the other hand, the development of digital infrastructure makes it more convenient for returnee entrepreneurs to access the knowledge and resources they require [28], enhancing the perceived feasibility of entrepreneurship to some extent, thus lowering entry barriers and aiding in the development of high-quality entrepreneurial activities.

## 2.3. Navigating the complex path and mechanism of institutional environment configuration to promote returnee entrepreneurial quality: A configuration perspective

Complex systems theory posits that in composite systems, components interact complexly, not simply additively or sub-tractively, forming intricate internal structures. Single or combined institutional elements may not directly boost returnee entrepreneurship quality; their interaction within the institutional environment configuration is required for effectiveness [59]. This paper examines the configurational effects among institutional environmental elements to uncover the complex relationship between institutional configurations and returnee entrepreneurship quality.

Firstly, institutional complexity creates varied combinatory effects among environmental elements. For example, Sendra-Pons et al. [60] examined six institutional elements' complex impact on entrepreneurship rates, finding two configurational pathways to boost national entrepreneurship rates.

Secondly, the interplay of reasonable entrepreneurial resource allocation, reduced institutional barriers, enhanced organizational legitimacy, and lower transaction costs determines returnee entrepreneurship quality. The institutional environment can lower barriers and costs for returnee firms via resource allocation. Unfair industrial policies often cause barriers, creating an unfair competition that favors established firms and hampers new ventures [38]. Reasonable resource allocation also steers firms toward R&D innovation over rent – seeking [36]. Enhanced legitimacy reduces barriers, as some

scholars note entrepreneurs' risk perception influences barriers, and returnees' improved legitimacy lowers these barriers [15]. Thus, optimizing the institutional environment must consider the synergy of barriers, resource allocation, legitimacy, and costs.

Finally, different regions may develop distinct institutional environmental ecologies, resulting in varied pathways to enhance returnee entrepreneurship quality. Economically advanced regions can optimize formal and informal institutional environments to lower barriers and raise the legitimacy threshold for returnee firms. This synergy creates opportunities, resources, and legitimacy, improving entrepreneurship quality. In less economically developed regions, optimizing institutional environments can cut returnee firms' transaction costs, spurring innovation investment and supporting quality company development. The paper, considering institutional isomorphism sociologically and transaction costs economically, integrates four driving mechanisms and their institutional sources, forming a multifaceted model to boost returnee entrepreneurship quality.

In summary, this paper adopts a configurational perspective to explore the configurational effects among institutional environmental elements, revealing the complex driving mechanisms for achieving high-quality returnee entrepreneurship. The theoretical foundation and research framework are illustrated in Fig 1.

## 3. Methodology

### 3.1. Method

The QCA tool can qualitatively assess whether individual institutional factors are necessary for enhancing the quality of returnee entrepreneurship. However, it fails to quantitatively describe the degree of necessity, especially given that fuzzy sets encompass meanings that transcend mere "yes or no" responses [4]. Therefore, combining the methods of NCA and QCA offers greater utility. First of all, NCA is employed to test for necessary conditions. It explores whether a certain level of any individual or institutional factor constitutes a bottleneck condition for promoting the quality of returnee entrepreneurship. In addition, fsQCA is used to verify the necessity analysis of individual factors. Finally, considering the dynamic nature of antecedent conditions, a temporal component is introduced into QCA, with the stages of returnee entrepreneurship development serving as the basis for dividing periods. Specifically, 2014–2016 is categorized as Period 1, representing the initial development stage of returnee entrepreneurship in China. What is more, 2017–2019 is categorized as Period 2, marked by the policy-driven and rapid growth of returnee entrepreneurship. Finally, 2020–2022 is categorized as Period 3, and is characterized by the stabilization and diversification of returnee entrepreneurship. Based on this framework, the evolution of conditions and configurations over time is explored, along with how and why the provinces and cities corresponding to these configurations have changed.

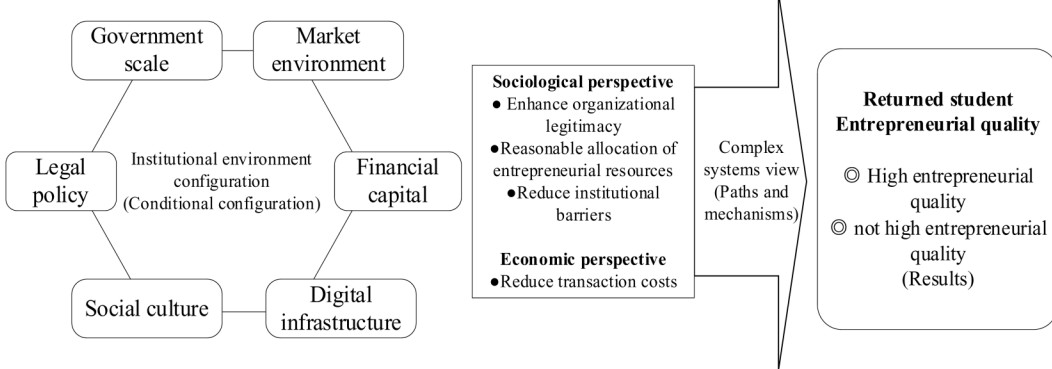

**Fig 1. Theoretical basis and research framework.**

## 3.2. Data collection

This study primarily uses 30 provinces in China (excluding Hong Kong, Macao, and Taiwan) as research samples. Due to data limitations, Qinghai Province was excluded.

The data on government size is sourced from the "China Statistical Yearbook". Data on the market environment comes from the "China Provincial Marketization Index Report (2018)" and "China Provincial Marketization Index Report (2021)". Financial capital data is obtained from the "China Statistical Yearbook" and the statistical yearbooks of each province. Legal and policy data is sourced from the National Intellectual Property Administration. Sociocultural data is collected from the "China Civil Affairs Statistical Yearbook", ICE8000, and the "China Statistical Yearbook". Data on digital infrastructure is sourced from the "China Information Yearbook" and the "China Statistical Yearbook". Data on the quality of returnee entrepreneurship is primarily sourced from the CSMAR database. The study sets the outcome variable one year after the conditions, meaning that the six antecedent conditions use data from 2013 to 2021. The quality of returnee entrepreneurship is based on data from 2014 to 2022.

## 3.3. Measurement

To effectively eliminate the impact of original indicators' dimensions during the measurement phase, we first normalized the raw data. In this study, normalization was performed using SPSS software via the Min-Max normalization method, which standardizes data of varying dimensions by mapping it to the [0, 1] interval. Compared to Z-score normalization (which may yield negative values), the [0, 1] interval better aligns with the set – membership logic of QCA. The formula is:

$$X_{normal} = \frac{X - X_{min}}{X_{max} - X_{min}}$$

where X is the raw data, $X_{min}$ and $X_{max}$ are the minimum and maximum values of the dataset, and $X_{normal}$ is the normalized value. Descriptive statistics of the normalized data are presented in Table 1. Measurement indicators for the six institutional environment factors are as follows:

**3.3.1. Government scale.** The scale of government is generally measured by public fiscal expenditure. Following Li et al. [61], this paper measures it based on the ratio of general government budget expenditure to GDP.

**3.3.2. Market environment.** Drawing on the methodology of Chen et al. [62], this study employs the marketization index to measure the market environment across different regions. The marketization index is a comprehensive indicator that reflects the degree of marketization in a region's economy. It typically encompasses various characteristics such as the relationship between the government and the market, the development of a non-state-owned economy, the degree of product market development, and the development of factor markets. By utilizing this index, the study aims to capture the nuances of the market environment in different regions, which in turn may influence the quality of returnee entrepreneurship.

**Table 1. Calibration and descriptive statistics.**

| Conditions | Fuzzy set calibrations | | | Measure descriptive | | | |
|---|---|---|---|---|---|---|---|
| | Fully in | Fully in | Fully out | Mean | SD | Max | Min |
| Returnee entrepreneurial quality | 0.073 | 0.026 | 0.007 | 0.082 | 0.148 | 1.000 | 0.000 |
| Government scale | 0.171 | 0.102 | 0.070 | 0.141 | 0.156 | 1.000 | 0.000 |
| Market environment | 0.735 | 0.654 | 0.538 | 0.631 | 0.158 | 1.000 | 0.000 |
| Financial capital | 0.103 | 0.063 | 0.039 | 0.107 | 0.153 | 1.000 | 0.000 |
| Legal policy | 0.180 | 0.109 | 0.053 | 0.156 | 0.168 | 1.000 | 0.002 |
| Social culture | 0.441 | 0.361 | 0.239 | 0.340 | 0.161 | 0.824 | 0.000 |
| Digital infrastructure | 0.258 | 0.189 | 0.118 | 0.210 | 0.137 | 0.939 | 0.037 |

**3.3.3. Financial capital.** Inspired by Stuetzer et al. [63], this paper measures the level of financial capital in each region. To achieve this it is using the ratio of financial institution deposit balances to the permanent population.

**3.3.4. Legal policy.** Legal policies primarily reflect policy transparency and judicial fairness within a region. Drawing on the work of Shen and Huang [64], this paper quantifies the level of intellectual property protection. This is achieved by calculating the ratio of the number of intellectual property cases to the regional GDP share, serving as a measure of the legal and policy environment in each region.

**3.3.5. Sociocultural context.** The social and cultural environment mainly represent the social trust and tolerance of the returnees in a region. In this paper, ICE8000's provincial-level integrity ranking was used to measure regional social trust. The formula for calculating regional social trust is as follows:

$$TR_{it} = \left[ \frac{1}{30} \times (31 - ICE) \right] \times \frac{1}{30} \sum_{i=1}^{30} CT_{it}$$

Where "i" represents the region, "t" represents the year, "TR" represents the degree of regional social trust. What is more, "ICE" represents the annual provincial integrity rankings of China's regions published by ICE8000, and "CT" represents the turnover of commodity trading markets with transactions exceeding 100 million yuan. Social organizations play a significant role in the subjectivity of the sociocultural environment and can address issues of social trust [65]. Following that, this paper uses the ratio of the number of social organizations to the permanent resident population to measure the inclusiveness of the region. The total score for the sociocultural environment component is determined by calculating the weighted sum of each indicator through principal component analysis.

**3.3.6. Digital infrastructure.** Following Chen et al. [66], the level of digital infrastructure in each region is measured using a weighted score of the number of internet users per hundred people, the proportion of computer service and software employees, and the number of mobile phone users per hundred people. The weights are determined through principal component analysis.

**3.3.7. Regional returnee entrepreneurial quality.** Overseas returnee entrepreneurship has become a key factor driving high-quality economic development in China, characterized by high growth and innovation [8]. Existing research measures the quality of urban entrepreneurship by the number of companies listed on the Growth Enterprise Market (GEM). This reflects the growth of high-quality entrepreneurship [67]. Overseas returnee entrepreneurs often favor high-quality entrepreneurship by leveraging new technologies and business models acquired overseas [5]. Companies listed on the Science and Technology Innovation Board (STAR Market) exemplify the use of high technology by overseas returnees, showcasing the innovation dimension of high-quality entrepreneurship. Therefore, this paper integrates data from both the GEM and STAR markets to measure the quality of overseas returnee entrepreneurship in the region. This is primarily due to the following three reasons.

First of all, the objectivity of market data is impotant. We choose the GEM and STAR markets as platforms to measure the quality of overseas returnee entrepreneurship. That is because these markets have strict listing standards and regulatory requirements. This way, they enable the screening of companies with high growth potential and innovative capabilities. The number of listed companies, as a measurement indicator, is objective and verifiable, directly reflecting the activity and market recognition of overseas returnee entrepreneurial activities in the region.

Moreover, the aim is to reflect the characteristics of innovation and high growth. The GEM and STAR markets particularly focus on the innovative capabilities and growth potential of startups. This aligns well with the typical characteristics of overseas returnee entrepreneurship (such as the application of new technologies and business model innovation). The number of companies listed on these markets can indirectly reflect the contributions of overseas returnee entrepreneurship in terms of innovation and high growth.

Furthermore, the basis is the data availability and representativeness. Listed company data is easily accessible and publicly transparent, facilitating cross-regional and cross-temporal comparisons and analyses. Companies in these markets often represent leaders and innovators in their industries, thus having high representativeness. In summary, although a single indicator (such as the number of listed companies) may not fully cover entrepreneurship quality (such as innovation, sustainability, job creation, etc.), it is one of the currently quantifiable and more direct indicators reflecting the quality and effectiveness of entrepreneurial activities. The listing standards of the GEM and STAR markets encompass considerations of multiple qualities such as corporate innovation capabilities, financial status, and market prospects. Therefore, this indicator can, to a certain extent, comprehensively reflect multiple dimensions of entrepreneurship quality.

Following Xie et al. [67], this measurement specifically adopts the total number of companies listed on the GEM and STAR markets divided by the total permanent resident population. The selection criteria include selecting listed companies whose founders or executive teams have overseas backgrounds. During the selection process, ST companies and samples with missing or abnormal data are excluded.

### 3.4. Calibration

Calibration, a process of assigning set – membership scores to cases, adopts the direct method using the calibrate function in fsQCA software. Drawing on prior research [68], the three calibration points for each variable are set at the upper quartile (full membership threshold), median (crossover point threshold), and lower quartile (full non-membership threshold) of the sample data. To prevent ambiguous 0.5 membership scores that hinder case classification and analysis, a 0.001 upward adjustment is made to scores below 1. The calibration results are shown in Table 1.

## 4. Results

### 4.1. Necessary condition analysis

In the Necessary Condition Analysis, the CE-FDH and CR-FDH ceiling lines are applicable for most uses of NCA. It is common in research to use both default ceiling lines simultaneously for analysis, which aids in stability testing by comparing results [69]. Therefore, this paper derives effect sizes using both CE and CR estimation methods. The blank space acceptance indicates that the necessary condition relationship needs to meet the following two necessary, but not sufficient, conditions: the effect size is greater than the selected threshold ($d > 0.1$), and the p-value of the effect size is less than the selected threshold ($p < 0.05$), meaning the effect size is significant. The analysis results, as shown in Table 2, indicate that the effect sizes of market environment, financial services, sociocultural factors, and digital infrastructure are less than 0.1, with p-values greater than 0.05. This suggests that these four institutional factors do not constitute necessary conditions for influencing the quality of overseas returnee entrepreneurship. In contrast, the effect sizes of government size and legal policies are both greater than 0.1, with significant p-values. Therefore, a certain level of government size and legal policies is a necessary condition for achieving a certain level of quality in overseas returnee entrepreneurship.

The bottleneck table allows for the analysis of the necessary level (%) of antecedent conditions (minimum requirement level) for a given Y value of the outcome. It also permits the expression of hypotheses in terms of degree levels [69]. The analysis results presented in Table 3 indicate that when the quality of overseas returnee entrepreneurship is below 10%, the size of the government is not a necessary condition (NN stands for "not necessary") for this outcome. However, when the quality of overseas returnee entrepreneurship reaches 70%, the following conditions are required: 2% of government size, 100% of market environment, 17.8% of financial capital, 3.1% of legal policies, 4.3% of sociocultural factors, 32.6% of digital infrastructure.

### 4.2. Necessity analysis of individual elements by fsQCA

Before conducting a sufficiency analysis, it is necessary to explore whether a certain condition is necessary for the occurrence of a given result, meaning whether it must be present or absent for the result to occur [70]. To this end, this paper

further employs the fsQCA method to analyze the necessary conditions of individual elements, looking for conditions with coverage and consistency thresholds greater than 0.9. The data presented in Table 4 reveals that, for both Period 1 and Period 2, the consistency values of all six antecedent conditions fall below the threshold of 0.9, indicating the absence of a solitary institutional environment factor that is indispensable for attaining high-quality entrepreneurship among overseas returnees. Conversely, during Period 3, the consistency levels of both the market environment and digital infrastructure surpass the 0.9 mark. This highlights their pivotal roles as critical institutional elements in advancing the quality of overseas returnee entrepreneurship between 2020 and 2022. Consequently, in conducting the fsQCA sufficiency analysis for Periods 1 and 2, we refrain from incorporating any necessary conditions. However, for the fsQCA sufficiency analysis of Period 3, we will specifically incorporate the market environment and digital infrastructure as necessary conditions for our analysis.

**Table 2. Necessary condition analysis results.**

| Conditions[a] | Method | Accuracy | Scale | Ceiling zone | Effect size[b] | P-value [c] |
|---|---|---|---|---|---|---|
| Government Scale | CE | 100% | 1 | 0.020 | 0.917 | 0.005 |
| | CR | 99.6% | 1 | 0.013 | 0.940 | 0.005 |
| Market environment | CE | 100% | 1 | 0.771 | 0.000 | 0.000 |
| | CR | 92.6% | 1 | 0.795 | 0.000 | 0.000 |
| Financial capital | CE | 100% | 1 | 0.140 | 0.004 | 0.001 |
| | CR | 91.5% | 1 | 0.131 | 0.042 | 0.004 |
| Legal policy | CE | 100% | 0.998 | 0.024 | 0.772 | 0.008 |
| | CR | 99.6% | 0.998 | 0.021 | 0.766 | 0.008 |
| Social culture | CE | 100% | 8 | 0.254 | 0.028 | 0.003 |
| | CR | 98.9% | 8 | 0.236 | 0.028 | 0.003 |
| Digital infrastructure | CE | 100% | 0.902 | 0.232 | 0.000 | 0.000 |
| | CR | 100% | 1 | 0.020 | 0.917 | 0.005 |

Note: a. Membership value of calibrated fuzzy set. b. 0.0 ≤ d < 0.1: a "small effect"; 0. 1 ≤ 1 d < 0 3: a"Medium effect"; d ≥ 0.3: a" large effect". c. Substitution testing in NCA analysis (permutation test, Number of redraws = 10000)

**Table 3. NCA bottleneck level (%) analysis results.**

| Returnee entrepreneurial quality | Government scale | Market environment | Financial capital | Legal policy | Social culture | Digital infrastructure |
|---|---|---|---|---|---|---|
| 0 | NN | 35.5 | 1.5 | NN | NN | 3.7 |
| 10 | NN | 45.7 | 3.8 | 0.2 | 0.2 | 7.8 |
| 20 | 0 | 55.8 | 6.2 | 0.7 | 0.9 | 12 |
| 30 | 0.5 | 65.9 | 8.5 | 1.2 | 1.6 | 16.1 |
| 40 | 0.9 | 76.0 | 10.8 | 1.7 | 2.3 | 20.2 |
| 50 | 1.3 | 86.1 | 13.1 | 2.1 | 2.9 | 24.4 |
| 60 | 1.7 | 96.2 | 15.4 | 2.6 | 3.6 | 28.5 |
| 70 | 2 | NA | 17.8 | 3.1 | 4.3 | 32.6 |
| 80 | 2.4 | NA | 20.1 | 3.6 | 5 | 36.8 |
| 90 | 2.8 | NA | 22.4 | 4 | 5.6 | 40.9 |
| 100 | 3.2 | NA | 24.7 | 4.5 | 6.3 | 45 |

## 4.3. Multi-period sufficiency configuration analysis

When the consistency check value exceeds the measurement standard, it is indicated that the antecedent condition is a sufficient condition for the outcome. Referring to existing research [71], this paper sets the consistency threshold at 0.8 and the frequency threshold at 1 and uses an RPI consistency threshold greater than 0.8 as the screening criterion. In this paper, the multi-period QCA method in the dynamic QCA method is used to analyze the institutional configuration leading to high-quality overseas returnees' entrepreneurship, and the results are shown in Table 5. Following Fiss [68], "•" indicates the presence of a condition, and its absence is indicated by "⊗". A large circle represents the factor as a core condition, a small circle as a peripheral condition, and a blank space indicates that the condition is not critical in the configuration.

### 4.3.1. Configurations that drive high- quality returnee entrepreneurship. *Configuration Analysis for Period 1*.
Resource-Driven under Opportunity Emergence. Configuration H1 suggests that an institutional environment with high financial capital and a moderate government size serves as core conditions for high-quality entrepreneurship among overseas returnees. This configuration also includes peripheral conditions. These peripheral conditions consist of a high

**Table 4. The necessary condition analysis of individual factors in each period.**

| | Period 1 | | Period 2 | | Period 3 | |
|---|---|---|---|---|---|---|
| | Consistency | Coverage | Consistency | Coverage | Consistency | Coverage |
| Government Scale | 0.263 | 0.201 | 0.321 | 0.287 | 0.243 | 0.301 |
| ~Government Scale | 0.893 | 0.726 | 0.828 | 0.829 | 0.841 | 0.903 |
| Market environment | 0.857 | 0.779 | 0.874 | 0.852 | 0.917 | 0.920 |
| ~Market environment | 0.343 | 0.238 | 0.296 | 0.272 | 0.168 | 0.227 |
| Financial capital | 0.550 | 0.823 | 0.683 | 0.671 | 0.861 | 0.683 |
| ~Financial capital | 0.536 | 0.287 | 0.435 | 0.397 | 0.229 | 0.481 |
| Legal policy | 0.661 | 0.613 | 0.741 | 0.525 | 0.757 | 0.626 |
| ~Legal policy | 0.537 | 0.367 | 0.354 | 0.502 | 0.317 | 0.602 |
| Social culture | 0.529 | 0.513 | 0.710 | 0.645 | 0.764 | 0.712 |
| ~Social culture | 0.654 | 0.434 | 0.424 | 0.417 | 0.305 | 0.459 |
| Digital infrastructure | 0.411 | 0.966 | 0.726 | 0.742 | 0.951 | 0.617 |
| ~Digital infrastructure | 0.686 | 0.324 | 0.421 | 0.370 | 0.122 | 0.618 |

**Table 5. The configuration of high returnee entrepreneurship is generated.**

| Conditional configuration | Period 1 | | | Period 2 | | | Period 3 | | |
|---|---|---|---|---|---|---|---|---|---|
| | H1a | H1b | H1c | S1 | S2a | S2b | P1a | P1b | P1c |
| Government Scale | ⊗ | ⊗ | ⊗ | ⊗ | ⊗ | ⊗ | ⊗ | ⊗ | |
| Market environment | • | • | • | • | ● | ● | ● | ● | ● |
| Financial capital | ● | ● | ● | ⊗ | • | • | • | | • |
| Legal policy | | • | ⊗ | • | • | ⊗ | | • | |
| Social culture | ⊗ | | • | ● | • | ⊗ | | | • |
| Digital infrastructure | • | • | ⊗ | ⊗ | ● | ● | ● | ● | ● |
| Consistency | 0.997 | 0.997 | 0.94 | 0.971 | 0.966 | 0.875 | 0.942 | 0.933 | 0.973 |
| Raw coverage | 0.283 | 0.299 | 0.159 | 0.236 | 0.422 | 0.153 | 0.716 | 0.631 | 0.642 |
| Unique coverage | 0.015 | 0.032 | 0.038 | 0.126 | 0.254 | 0.054 | 0.015 | 0.058 | 0.054 |
| Overall coverage | 0.376 | | | 0.602 | | | 0.829 | | |
| Overall consistency | 0.974 | | | 0.965 | | | 0.938 | | |

market environment combined with high digital infrastructure (H1a), robust legal policies (H1b), or strong sociocultural factors (H1c). This configuration suggests that, within an institutional framework where financial services and the market environment are both favorable, overseas returnee entrepreneurial ventures significantly benefit from abundant financial capital within the region, easily obtaining the necessary funding support during their initial stages and growth processes. Additionally, the market environment is filled with rich entrepreneurial opportunities and resources, providing fertile soil for these ventures to achieve high-quality entrepreneurial goals. As shown in Table 5, this type of institutional environment configuration is named the Resource-Driven under Opportunity Emergence model. Typical provinces that fit this configuration include leading provinces such as Guangdong, Jiangsu, Beijing, and Shanghai. Taking Shanghai as an example, it is one of the regions with the highest concentration of high-quality entrepreneurial ventures by overseas returnees in China. First of all, Shanghai has issued a series of policy measures to support and encourage high-quality entrepreneurship by overseas returnees. For example, by providing startup funding and entrepreneurial training, which reduce the costs and risks of starting a business for overseas returnees. Additionally, Shanghai has continuously strengthened the development of its social credit system, creating a social atmosphere of fair competition, integrity, and inclusiveness. Furthermore, Shanghai has promoted the construction of a city blockchain digital infrastructure system project, providing overseas returnee entrepreneurs with secure, trustworthy, efficient, and convenient blockchain services. In summary, Shanghai actively fosters a business environment that is legal, convenient, and international. This environment provides ample entrepreneurial opportunities and resources for high-quality entrepreneurship among overseas returnees. Additionally, it strengthens the legitimacy of these ventures within the market and society, thereby enhancing the overall quality of entrepreneurship. This aligns with the characteristics of the institutional environment configuration featuring the opportunity-resource synergistic legitimacy drive presented in this paper.

*Configuration Analysis for Period 2*. Legitimacy-Driven by Emerging Opportunities. Configuration S1 indicates that a combination of high sociocultural factors and a moderate government scale serves as core conditions for high-quality overseas returnee entrepreneurship. Additionally, it includes peripheral conditions such as a high market environment, moderate financial services, strong legal and policy support, and moderate digital infrastructure. Together, these elements create an institutional environment that fosters high-quality entrepreneurship among overseas returnees. This configuration indicates that overseas returnee startups primarily leverage a favorable market environment and sociocultural factors. This is particularly true in contexts with inadequate financial services, limited digital infrastructure, and a shrinking government scale. By utilizing these strengths, returnee startups can seize high-quality entrepreneurial opportunities and enhance organizational legitimacy, ultimately achieving high-quality entrepreneurial activities. The government plays a role as a "helping hand" by providing a robust property rights protection environment to ensure the smooth progress of corporate innovation activities. As shown in Table 5, this paper names such an institutional environment configuration as "Legitimacy-Driven by Emerging Opportunities". Typical provinces fitting this configuration include Anhui, Shandong, and Hubei. To begin with, and by taking Anhui Province as an example, the province actively integrates into the integrated development of the Yangtze River Delta. It also strengthens cooperation with Jiangsu, Shanghai, Zhejiang, and other regions to stimulate market vitality. Therefore, it provides high-quality entrepreneurial opportunities for overseas returnees. Furthermore, Anhui Province actively promotes the sharing and application of credit information. Its social trust index is higher than the national average. It also advances the construction of urban and rural community service systems, encouraging the development of social organizations. Thus, it creates an honest and inclusive sociocultural environment that enhances the legitimacy of overseas returnee startups. What is more, Anhui Province emphasizes governing according to the law, fostering a fair and just legal environment. This approach provides strong legal protection for overseas returnee startups and encourages their entrepreneurial spirit. Additionally, by optimizing the market environment to create more high-quality entrepreneurial opportunities, Anhui Province promotes trust and inclusiveness within the sociocultural fabric. This, in turn, enhances organizational legitimacy and supports the development of a service-oriented government. Ultimately, these efforts stimulate high-quality entrepreneurship among overseas returnees, aligning with the "Legitimacy-Driven by Emerging Opportunities" institutional environment configuration discussed in this paper.

Dual-Drive Model of Opportunities and Resources. Configuration S2 points out that an institutional environment configuration with a high market environment, high digital infrastructure, and medium government scale serves as core conditions for high-quality overseas returnee entrepreneurship. Additionally, this configuration includes peripheral conditions such as high financial services, strong legal and policy support, and significant sociocultural factors, whether high or moderate. Together, these elements contribute to fostering high-quality entrepreneurship among overseas returnees. Similar to Configuration H1, Configuration S2 relies on abundant entrepreneurial opportunities in the market, supported by effective digital infrastructure and financial services, to enhance the quality of overseas returnee entrepreneurship. However, unlike Configuration H1, the market environment in Configuration S2 transitions from a peripheral condition to a core condition. This shift indicates that improving the quality of overseas returnee entrepreneurship in this configuration requires a "dual-strong" driving mechanism, where opportunities and resources coexist and integrate within the region. As shown in Table 5, this paper names this institutional environment configuration under the "Dual-Drive Model of Opportunities and Resources". Typical provinces fitting this configuration include Beijing, Shanghai, Zhejiang, and Fujian. Taking Beijing as an example, the city boasts one of the most vibrant and mature market environments in China. This dynamic atmosphere attracts a substantial influx of domestic and overseas capital, nurturing numerous innovative companies and projects. As a result, it offers overseas returnee entrepreneurs a vast stage and countless opportunities. In this environment, every spark of innovation has the potential to rapidly transform into popular products or services, thus realizing commercial value.

Additionally, Beijing is a global leader in digital infrastructure development. With high-speed networks, intelligent urban management, and widespread applications of cutting-edge technologies such as cloud computing and big data, the city provides robust technical support and convenient conditions for overseas returnee entrepreneurs. These technologies not only lower the barriers to entry for entrepreneurship but also greatly improve entrepreneurial efficiency and market response speed, enabling entrepreneurs to adapt more quickly to market changes. In terms of financial services, Beijing is home to numerous well-known domestic and overseas financial institutions and venture capital companies. Thus it forms a multi-tiered and diversified financial service system. This provides overseas returnee entrepreneurs with sufficient funding sources, whether it be angel investments during the startup phase, venture capital during the growth phase, or IPO listings during the mature phase, all of which can be found here. Although legal and policy support and the sociocultural environment are not core conditions in Configuration S2, Beijing excels in both areas. The government has continuously optimized the business environment and issued a series of policies and measures to encourage innovation and entrepreneurship, providing good policy support and legal guarantees for overseas returnee entrepreneurs. At the same time, Beijing's open and inclusive sociocultural atmosphere promotes exchanges and cooperation among entrepreneurs from different cultural backgrounds. That way it can stimulate further its entrepreneurial vitality.

In summary, Beijing exemplifies the Dual-Drive Model of Opportunities and Resources institutional environment configuration. Its superior market environment, advanced digital infrastructure, comprehensive financial service system, and open, inclusive sociocultural atmosphere create an ideal entrepreneurial ecosystem for overseas returnee entrepreneurs. This rich environment is abundant with opportunities and resources, effectively enhancing the quality of overseas returnee entrepreneurship.

***Configuration Analysis for Period 3.*** Dual – Driver Model of Opportunities and Resources. Configuration P1 shows an institutional environment with high market environments and digital infrastructure as core conditions, plus medium government scale with high financial services (P1a), medium government scale with high legal policies (P1b), or high financial environments with high sociocultural factors (P1c) as peripheral conditions, can drive high – quality overseas returnee entrepreneurship. Like Configuration S2, P1 also needs a dual – strong opportunities and resources mechanism to improve overseas returnee entrepreneurship quality. But unlike S2, the medium government scale in P1 changes from a core to a peripheral or irrelevant condition, indicating a smaller negative impact on international returnee entrepreneurship. Also, P1's resource driver is digital infrastructure, not financial resources. As Table 5 shows, this institutional configuration is called the dual – driver model of opportunities and resources. Provinces like Jiangsu, Zhejiang, Tianjin,

and Guangdong fit this pattern. Guangdong Province, for example, has a highly vibrant market environment. It's home to many well-known domestic and international companies, forming a comprehensive industrial and supply chain system. This gives overseas returnee entrepreneurs vast market space and rich business resources. Guangdong is also a leader in China's digital infrastructure. The widespread use of advanced tech like 5G, big data centers, and cloud computing platforms offers entrepreneurs strong technical support and convenient services. Guangdong also excels in financial services. It hosts many financial institutions and venture capital firms, creating a diversified financial service system. This offers international returnee entrepreneurs sufficient funding and diverse financing options. Whether it's startup – phase angel investment, growth-phase venture capital, or maturity-phase IPO listing, entrepreneurs can find suitable financing channels here. On legal policies, Guangdong's government keeps enhancing the business environment with policies and measures to encourage innovation and entrepreneurship. These provide good policy support and legal guarantees for international returnee entrepreneurs, lowering startup barriers, improving efficiency, and helping entrepreneurs adapt to market changes quickly to seize opportunities. Moreover, Guangdong's sociocultural environment is highly appealing. It's open, inclusive, innovation-respecting, and competition-encouraging, with a strong innovation and entrepreneurship atmosphere. Entrepreneurs from diverse cultural backgrounds can learn from each other here, jointly promoting innovation and entrepreneurship development. In summary, Guangdong Province exemplifies the dual-driver model of opportunities and resources in an institutional environment. With its superior market environment, advanced digital infrastructure, and comprehensive financial service system, it offers an ideal entrepreneurial ecosystem for overseas returnee entrepreneurs. The favorable legal and policy framework and open, inclusive sociocultural environment further enrich this landscape, providing abundant opportunities and resources.

## 4.4. Discussion

### 4.4.1. Evolutionary analysis of regional institutional configurations across multiple periods.

First of all, by carefully examining each condition (row) in Table 5, we can observe the changes in individual institutional elements within configurations across different periods. It is equally noteworthy that the element of government size consistently exhibits a trend of harming entrepreneurship by overseas returnees across the three consecutive periods. This finding reveals a crucial point: for entrepreneurship by overseas returnees. Further government financial support is not necessarily better. In addition, excessive intervention or support may instead have counterproductive effects, posing certain obstacles to entrepreneurial activities.

What is more, the market environment plays a pivotal role across all three time periods, and it has even ascended to a core condition in the last two time periods. This trajectory indicates that the market environment has always played a stable and positive role in enhancing the quality of entrepreneurship by overseas returnees. This positive effect has become increasingly significant over time, becoming an indispensable key factor for entrepreneurial success.

Furthermore, the importance of digital infrastructure begins to emerge gradually in Period 2 and rises to a core condition in Period 3, indicating a particularly striking change. Considering that Period 3 coincides with the COVID-19 pandemic, this change undoubtedly highlights the accelerating effect of the pandemic on the development of digital infrastructure and the increasingly critical role that digital infrastructure plays in entrepreneurial activities by overseas returnees. Catalyzed by the pandemic, the importance of digital infrastructure has been greatly elevated, providing overseas returnee entrepreneurs with a more convenient and efficient entrepreneurial environment and support.

By observing the columns (configurations) in Table 5, we can discern the evolutionary trajectories and patterns of configurations across different periods. In Period 1, the primary driving mechanism for the quality of entrepreneurship by overseas returnees is resource-driven, with the emergence of opportunities, primarily fueled by financial resources and supplemented by the emergence of opportunities in the market environment. In Period 2, Configuration H1 evolves in two directions. Firstly, in the context of opportunity emergence, the resource-driven mechanism of financial capital shifts to a legitimacy-driven mechanism rooted in social culture. This transition evolves the resource-driven

type under opportunity emergence in Configuration H1 into the legitimacy-driven type under opportunity emergence in Configuration S1. Secondly, while the resource-driven scenario remains unchanged, the role of the market environment becomes more prominent. Here, opportunity emergence transforms into an opportunity-driven approach, evolving the resource-driven type under opportunity emergence in Configuration H1 into the dual-drive model of opportunities and resources in Configuration S2. In Period 3, the driving effect of Configuration S1 weakens, while the driving effect of Configuration S2 continues to play a role. For example, the institutional configuration of the dual-drive type of opportunity and resource (S2, P1) maintains a relatively stable effect in achieving high-quality entrepreneurship by overseas returnees.

### 4.4.2. Differentiated paths for enhancing the quality of entrepreneurship by overseas returnees in regions with varying levels of economic development.

Due to the influence of domestic resource endowments and economic development levels, the institutional environments, and entrepreneurial activities of overseas returnees in various provinces of China exhibit significant heterogeneity. Based on differences in economic development levels, provinces are classified across different periods to analyze the differentiated paths for enhancing the quality of entrepreneurship by overseas returnees in different regions. As shown in Table 6. According to the level of economic development, China can be divided into:

Eastern Regions: Including 11 provinces, autonomous regions, and municipalities directly under the central government such as Beijing, Tianjin, Hebei, Liaoning, Shanghai, Jiangsu, Zhejiang, Fujian, Shandong, Guangdong, and Hainan. These regions have the highest level of economic development, primarily focusing on high-tech industries and modern service industries. They have a high degree of openness to the outside world and, a strong ability to attract overseas investment, and their economic aggregate and per capita income are both above the national average.

Central Regions: Including eight provinces and autonomous regions such as Shanxi, Jilin, Heilongjiang, Anhui, Jiangxi, Henan, Hubei, and Hunan. These regions have a secondary level of economic development, primarily focusing on heavy industry and agriculture. Geographically, they serve as a bridge between the East and the West, with convenient transportation, and have gradually become important manufacturing and logistics centers nationwide.

Western Regions: Including twelve provinces and autonomous regions such as Chongqing, Sichuan, Guizhou, Yunnan, Tibet, Shaanxi, Gansu, Ningxia, Qinghai, Xinjiang, Guangxi, and Inner Mongolia. These regions have a relatively low level of economic development, primarily focusing on resource development and traditional agriculture. In recent years, infrastructure construction and development efforts have been intensified, with enormous potential for economic development.

In Period 1, Eastern regions prioritized market environments, financial services, and digital infrastructure development, but varied in legal policies, sociocultural traits, and government scale. Central and Western regions showed no standout performance.

In Period 2, Eastern regions' institutional configurations diversified but still prioritized market environments and digital infrastructure. Central regions focused on market environments and legal policies, stressing sociocultural development but lacking financial services and digital infrastructure. Western regions still had no remarkable performance.

In Period 3, Eastern regions universally sought high – quality market environments, financial services, and digital infrastructure, with some configurations highlighting legal policies and sociocultural traits. Central regions also aimed for high–quality market environments, financial services, and digital infrastructure, with some configurations emphasizing legal policies and sociocultural traits. Western regions mainly pursued high – quality market environments, financial services, and digital infrastructure, with some configurations focusing on sociocultural development.

Overall, regions' development strategies across periods showed both differences and consistencies. Eastern regions' success in market environments, financial services, and digital infrastructure is worth other regions adopting. Central regions should bolster financial services and digital infrastructure. Western regions should focus on sociocultural development based on market environments, financial services, and digital infrastructure. Through learning, adapting, developing, and cooperating, regions can create more complete and competitive institutional configurations.

**Table 6. Membership table of institutional configurations for various provinces and cities at different time periods.**

| | | Configuration feature | Representative province |
|---|---|---|---|
| Period 1 (2014–2016) | H1a | **Non-high government size**\*High market environment\***High financial Services**\*Non-high sociocultural\*High digital infrastructure | **Eastern region**:Beijing,shanghai,Guangdong |
| | H1b | **Non-high governmentsize**\*High market environment\***High financial Services**\*Non-high legalpolicy\*High digital infrastructure | **Eastern region**:Shanghai,Beijing,zhejiang |
| | H1c | **Non-high governmentsize**\*High market Environment\***High financial services**\*Non-high Legalpolicy\*high socialculture\*Non High digital infrastructure | **Eastern Region**:Jiangsu |
| Period 2 (2017–2019) | S1 | **Non-high government size**\*High market Environment\*Non-high financial services\* High legal policy\***High social culture** Non-high digital infrastructure | **Eastern region**:Shandone **central region**:Anhui,Hube |
| | S2a | **Non-high goverment size**\*High market environment\* High financial Services\*High legalpolicy\*High social Cuture\***High digital infrastructure** | **Eastern region**: Beijing,Shanghai |
| | S2b | **Non-high goverment size**\*High market environment\* High financial Services\*High legalpolicy\*High social Cuture\***High digital infrastructure** | **Eastern region**:Jiangsu,Zhejiang,Fujian, Guangdong,Liaoning |
| Period 3 (2020–2022) | P1a | Non-high government size\***High market environment**\*High Financial services\* **High digital infrastructure** | **Eastern region:** Jiangsu,Zhejiang,Guangdong,Fujian Beijing,Shanghai,Shandong,Tianjin,Liaoning **Central region:** Hubei,Anhui **western region**:Sichuan,Shaanxi,chongqing |
| | P1b | Non-high government size\***High market Environment**\*High legal policy\***High digital infrastructure** | **Eastern region:**Guangdong Jiangsu,Shandong, shanghai, Beijing, Liaoning, Tianjin **Central region:** Hubei,Henan,Hunan **western region**:Sichuan,chongqing |
| | P1c | **High market environment**\*High financial Services\*High social culture\***High digital infrastructure** | **Eastern region:** Jiangsu,Zhejiang,Guangdong,Fujian,Shandong,Liaoning,Shanghai **Central region:**Hubei,Hebei,Anhui,Jiangxi **Western region:** Sichuan |

## 4.5. Sensitivity analysis

This paper employs three methods to validate the robustness of empirical results: varying the consistency threshold, changing the case frequency, and adjusting the calibration anchors [71]. Initially, the consistency threshold was raised from 0.8 to 0.85, which showed that the configurational paths generated were essentially consistent with the original paths. Subsequently, the case frequency was set to 2 for a reanalysis of configurations, and the results indicated that the configurational analysis with a case frequency of 2 was broadly similar to that with a case frequency of 1. Lastly, the

calibration anchors for full membership and non-membership were adjusted to the 95th and 5th percentiles, respectively, while the crossover point remained unchanged. The results demonstrated that the configurational paths obtained were encompassed within the original paths. These tests collectively confirm the robustness of the research findings in this study.

## 5. Conclusions and policy implications

This paper analyzes the quality of overseas returnees' entrepreneurship using data from 30 provinces in China, approaching the topic from the perspective of complex systems and configurations while employing the NCA and dynamic QCA method. It identifies multiple and intricate pathways through which institutional environment configurations enhance entrepreneurial quality.

Initially, the necessity analysis conducted through NCA reveals that a certain level of government size and legal policies is essential for achieving a desired quality level in overseas returnees' entrepreneurship. The fsQCA necessity analysis indicates that, in Periods 1 and 2, individual institutional elements are not necessary conditions for high-quality entrepreneurship. However, in Period 3, a high-quality market environment and robust digital infrastructure emerge as necessary conditions for fostering high-quality entrepreneurship.

Additionally, the enhancement of overseas returnees' entrepreneurship quality is driven by the synergistic effect of multiple factors. The effective combination of regional institutional elements enables high-quality entrepreneurship among overseas returnees, demonstrating that various pathways can lead to the same outcome.

In Period 1, there was one institutional configuration that could produce high-quality overseas returnees' entrepreneurship, namely, resource-driven with opportunity emergence. In Period 2, there were two institutional configurations capable of generating high-quality overseas returnees' entrepreneurship: legitimacy-driven with opportunity emergence and dual-wheel drive of opportunity and resources. In Period 3, there was one institutional configuration capable of producing high-quality overseas returnees' entrepreneurship, namely, the dual-wheel drive of opportunity and resources. For individual institutional elements, the market environment has played an important role throughout the period. The importance of digital infrastructure for the quality of overseas returnees' entrepreneurship has shifted from a marginal condition to a core one.

In terms of configurations, the market environment initially interacted closely with financial services but gradually began to strengthen its interaction with digital infrastructure over time. Finally, due to the different economic development levels and resource endowments among provinces, there are differentiated pathways for enhancing the quality of overseas returnees' entrepreneurship in regions with different economic development levels across different periods.

### 5.1. Theoretical contributions

Firstly, this paper establishes an integrated analytical framework for the quality of returnee entrepreneurship based on the institutional theory. Meanwhile, existing research often explores the development and growth patterns of returnee entrepreneurial firms from the perspectives of networks and resources at the micro-level [5], with mostly insufficient emphasis on returnee entrepreneurial activities at the regional level. Nonetheless, current studies focus on the pathways to enhance overall entrepreneurial quality, neglecting the unique group of returnee entrepreneurs whose pathways to quality enhancement differ from general entrepreneurship. This paper derives six conditions affecting the quality of returnee entrepreneurship from the special institutional context faced by returnees in emerging economies, constructing an integrated analytical framework for returnee entrepreneurship quality. This not only broadens the academic understanding of the macro-institutional context in which returnee entrepreneurship is embedded but also enriches the current research on entrepreneurial quality.

Secondly, the paper introduces a complex systems perspective, integrating institutional theories from both sociology and economics to dissect the complex impact mechanisms of institutional configurations on the quality of returnee

entrepreneurship. Existing research often focuses on the isomorphic effects of the institutional environment on regional entrepreneurial activities from a sociological perspective, overlooking the heterogeneity of institutions and the need for a more detailed analysis of how institutions affect returnee entrepreneurship from different research perspectives [23]. This paper responds to scholars' calls for effective dialogue between different research perspectives and disciplines in entrepreneurship research [13]. Based on a complex systems perspective, it analyzes how institutional configurations enhance the quality of returnee entrepreneurship through four impact mechanisms from two research perspectives, further focusing on the multiple driving pathways formed by the complex combination of these four mechanisms.

Fianlly, by employing the multi-period QCA method to examine the issue of scientific and technological talent agglomeration, this study responds to criticisms regarding the lack of consideration for the dynamic nature of antecedents. It also enriches the research methodologies within the field of entrepreneurial quality.Currently, existing research is primarily limited to quantitative studies that focus on linear relationships or qualitative studies centered on case analyses, often neglecting the impact of time on outcomes. This study elucidates how the coupling relationship between regional institutional factors and the quality of overseas returnees' entrepreneurship evolves, thereby contributing to the literature on dynamic QCA.

## 5.2. Practical and policy implications

From the macro-policy perspective, the Chinese government should consider several key actions.

To begin with, it should optimize its role and size. Research has shown that an overly large government can hinder overseas returnees' entrepreneurship. Therefore, the government needs to minimize unnecessary interventions and prioritize supportive policies and services, such as streamlining registration processes and offering tax incentives. Additionally, it should manage its size effectively to ensure efficient public service delivery while avoiding excessive burdens on market activities and entrepreneurial initiatives.

The government should also prioritize the enhancement of the market environment and digital infrastructure. Continuous improvements are necessary in the market, particularly in strengthening intellectual property protection and refining market access. These measures will create a fairer and more transparent marketplace, fostering better conditions for international returnees' entrepreneurship. It should also intensify investments in digital infrastructure, especially in central and western regions, to enhance its ability to attract and support overseas returnees' entrepreneurship.

It should also promote balanced regional development. The government should formulate differentiated regional development policies, encouraging eastern regions to continue leading while increasing support for central and western regions to improve their entrepreneurial environments. Through policy guidance and market mechanisms, it should promote cooperation and exchanges among regions to achieve resource-sharing and complementary advantages.

Secondly, with respect to provincial development tailored to local conditions, Eastern regions should continue leveraging their advantages in the market environment, financial services, and digital infrastructure. Their aim should be to attract more overseas returnees' entrepreneurial projects. At the same time, they should focus on legal policies and sociocultural development to provide more comprehensive institutional guarantees and a cultural atmosphere for overseas returnees' entrepreneurship. Central regions should enhance financial services and digital infrastructure to make their entrepreneurial environments more attractive.

Additionally, they should improve the market environment and legal policies to provide a more stable and predictable development environment for overseas returnees' entrepreneurship. Western regions should focus on sociocultural development on top of improving the market environment, financial services, and digital infrastructure, creating a more open and inclusive entrepreneurial atmosphere. Through policy guidance and financial support, they should further attract international returnees and entrepreneurial projects to drive rapid financial development in Western regions.

Lastly, from the perspective of overseas returnees' entrepreneurial companies, they should leverage institutional environmental advantages. Overseas returnees' entrepreneurial companies should thoroughly understand and utilize the

institutional environmental advantages of their locations, such as financial capital, market environment, and digital infra-structure, to create favorable conditions for their development.

They should enhance market sensitivity and innovation capabilities. These companies should maintain a keen sense of market dynamics, promptly seize entrepreneurial opportunities, and strengthen innovation capabilities to continuously improve the quality and competitiveness of their products and services.

They should focus on corporate culture and social responsibility. Overseas returnees' entrepreneurial companies should emphasize corporate culture, foster a positive working environment, and actively fulfill social responsibilities to establish a good corporate image, laying the foundation for sustainable development.

### 5.3. Limitations and future research directions

This study acknowledges certain limitations that future research should address. Firstly, this paper only examines the impact mechanisms of the macro-level institutional environment on the quality of returnee entrepreneurship. In reality, institutions are characterized by polycentricity and multilevel social structures, with returnee entrepreneurial firms embedded within multiple levels of institutions. Future research could explore the effects of interactions between different institutional levels on returnee entrepreneurship. Secondly, this paper adopts a hybrid approach combining NCA and dynamic QCA to investigate the macro-level impact of the institutional environment on the entrepreneurial quality of overseas returnees across various provinces in China. Simultaneously, it acknowledges a partial oversight of the micro-level influence of overseas returnees' institutional perceptions on their entrepreneurial choices. In future research, conducting interviews with returnee entrepreneurs and utilizing qualitative methods, such as case studies, could offer a more comprehensive understanding of how the institutional environment shapes the decisions of returnee entrepreneurs to pursue high-quality entrepreneurship. Finally, due to the limitations in existing research on measuring the entrepreneurial quality of overseas returnees, this paper focuses on the three characteristics of high-quality entrepreneurship: innovation, growth, and value creation. It employs the number of listed companies on the GEM and the Science and Technology Innovation Board as a measure of regional entrepreneurial quality. While this approach is both scientific and reasonable, it does not fully capture all three characteristics of high-quality entrepreneurship. Future research could improve upon this method to provide a more comprehensive assessment of the entrepreneurial quality of overseas returnees.

## Supporting information

**S1 Dataset. This is the S1 Dataset Title**.
(XLSX)

## Author contributions

**Conceptualization:** Yiyang Shen.

**Data curation:** Yiyang Shen.

**Funding acquisition:** Hongtao Yang.

**Investigation:** Yiyang Shen, Hongtao Yang.

**Methodology:** Yiyang Shen.

**Supervision:** Hongtao Yang, Qiuhua Zhu.

**Validation:** Yiyang Shen.

**Writing – original draft:** Yiyang Shen.

**Writing – review & editing:** Yiyang Shen, Qiuhua Zhu.

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
