## [Decision Letter · Decision Letter 0]

PONE-D-24-40215How does the institutional environment improve the entrepreneurial quality of returnees?

A configuration analysis based on a complex system viewPLOS ONE

Dear Dr. Yang,

Thank you for submitting your manuscript to PLOS ONE. After careful consideration, we feel that it has merit but does not fully meet PLOS ONE’s publication criteria as it currently stands. Therefore, we invite you to submit a revised version of the manuscript that addresses the points raised during the review process.

While this manuscript contributes original insights into the relationship between institutional configurations and returnee entrepreneurship in China using innovative methodologies (NCA and fsQCA), significant revisions are needed to strengthen its contributions. The literature review would benefit from integrating more recent studies from 2023 and 2024, especially in addressing the impact of the pandemic. The methodology presents limitations in sample size and selection, which affect the generalizability of the findings. Expanding the sample and incorporating longitudinal data would provide a more comprehensive analysis. Additionally, integrating qualitative insights and clarifying the operationalization of constructs are crucial for enhancing the study's robustness.

We look forward to receiving your revised manuscript.

Kind regards,

Farah Naz

Academic Editor

PLOS ONE

1. When submitting your revision, we need you to address these additional requirements. Please x`ensure that your manuscript meets PLOS ONE's style requirements, including those for file naming. The PLOS ONE style templates can be found at https://journals.plos.org/plosone/s/file?id=wjVg/PLOSOne_formatting_sample_main_body.pdf and https://journals.plos.org/plosone/s/file?id=ba62/PLOSOne_formatting_sample_title_authors_affiliations.pdf 2. Thank you for stating the following financial disclosure: "National Social Science Foundation of China�19BSH110��Huaqiao University's Academic Project Supported by the Fundamental Research Funds for the Central Universities�HQHRZX-202202�" Please state what role the funders took in the study.  If the funders had no role, please state: "The funders had no role in study design, data collection and analysis, decision to publish, or preparation of the manuscript." If this statement is not correct you must amend it as needed. Please include this amended Role of Funder statement in your cover letter; we will change the online submission form on your behalf. 3. We note that your Data Availability Statement is currently as follows: "All relevant data are within the manuscript and its Supporting Information files." Please confirm at this time whether or not your submission contains all raw data required to replicate the results of your study. Authors must share the “minimal data set” for their submission. PLOS defines the minimal data set to consist of the data required to replicate all study findings reported in the article, as well as related metadata and methods (https://journals.plos.org/plosone/s/data-availability#loc-minimal-data-set-definition). For example, authors should submit the following data: - The values behind the means, standard deviations and other measures reported;- The values used to build graphs;- The points extracted from images for analysis. Authors do not need to submit their entire data set if only a portion of the data was used in the reported study. If your submission does not contain these data, please either upload them as Supporting Information files or deposit them to a stable, public repository and provide us with the relevant URLs, DOIs, or accession numbers. For a list of recommended repositories, please see https://journals.plos.org/plosone/s/recommended-repositories. If there are ethical or legal restrictions on sharing a de-identified data set, please explain them in detail (e.g., data contain potentially sensitive information, data are owned by a third-party organization, etc.) and who has imposed them (e.g., an ethics committee). Please also provide contact information for a data access committee, ethics committee, or other institutional body to which data requests may be sent. If data are owned by a third party, please indicate how others may request data access.

Reviewers' comments:

Reviewer's Responses to Questions

**Comments to the Author**

1. Is the manuscript technically sound, and do the data support the conclusions?

Reviewer #1: Yes

Reviewer #2: Yes

2. Has the statistical analysis been performed appropriately and rigorously? 

Reviewer #1: Yes

Reviewer #2: N/A

3. Have the authors made all data underlying the findings in their manuscript fully available?

Reviewer #1: Yes

Reviewer #2: Yes

4. Is the manuscript presented in an intelligible fashion and written in standard English?

Reviewer #1: Yes

Reviewer #2: Yes

5. Review Comments to the Author

Reviewer #1: 1. Originality

The paper contributes original insights into the complex relationship between institutional configurations and returnee entrepreneurship in China. By integrating Necessary Condition Analysis (NCA) and fuzzy-set Qualitative Comparative Analysis (fsQCA), it approaches the subject from a configuration analysis perspective, which is relatively novel. However, further emphasis could be placed on how this study diverges from existing ones, especially by explicitly stating what gaps it fills in previous studies.

2. Literature Review

The literature review is thorough and aligns well with the research objectives. The authors draw on both sociological institutional theory and economic transaction cost theory, giving the review a solid theoretical foundation. However, integrating more recent literature (especially post-2020) could strengthen the relevance of the review. For instance, the study references literature from 2023 and 2024, but more exploration of how the pandemic impacted returnee entrepreneurship might add a contemporary dimension.

3. Latest Literature

The paper includes references from recent years, especially from 2021 and 2022, ensuring that it reflects contemporary debates in the field. However, the paper would benefit from a few more citations of highly relevant work published in 2023 to strengthen its claim of originality and engagement with the latest developments in institutional theory and entrepreneurship.

4. Methodology

The major weakness in this paper, according to the suggested methodology, lies in the sample size and selection. While the use of fsQCA and NCA methods is appropriate for analyzing complex institutional configurations, the selection of only 28 regions in China presents limitations:

Limited Generalizability

A larger sample size or inclusion of more diverse regions across China would enhance the robustness and external validity of the findings.

Selection Bias

The paper does not sufficiently explain whether selection bias was addressed. The regions analyzed might not be fully representative of all regions where returnee entrepreneurship occurs. This could mean that institutional configurations impacting other regions are overlooked, making the study's conclusions potentially less comprehensive.

Lack of Longitudinal Data

While the paper uses data from 2020 to 2022, entrepreneurship, especially returnee ventures, is often influenced by long-term institutional factors. The absence of a longitudinal approach (tracking changes over time) weakens the paper’s ability to capture evolving trends in how institutional environments impact entrepreneurship quality.

Considering returnee entrepreneurship is often a long-term process, using multi-year data could provide a deeper understanding of how institutional changes affect entrepreneurial success over time.

Over-reliance on Quantitative Methods

Although fsQCA and NCA provide strong quantitative insights into configurations, qualitative insights from interviews or case studies of returnee entrepreneurs might reveal nuanced factors that statistical methods miss. For example, first-hand accounts could shed light on how institutional support is experienced or perceived by returnees.

The paper could strengthen its methodology by integrating mixed methods—combining qualitative insights with the existing quantitative data to provide a more holistic view of the returnee entrepreneurial ecosystem.

Operationalization of Constructs

The operationalization of key constructs, such as entrepreneurial quality and institutional environment, might lack clarity. For instance, the study relies on metrics like the number of companies listed in certain markets, but this metric may not fully capture all dimensions of entrepreneurial quality (e.g., innovation, sustainability, or job creation).

The paper would benefit from a more detailed explanation of how these constructs are defined and measured, as well as any limitations in measurement.

Weakness in Addressing Regional Disparities

The paper mentions the uneven distribution of returnee entrepreneurship across regions but does not thoroughly explore how regional disparities in institutional environments affect the generalizability of its configurations. Certain configurations may only apply to more economically developed regions and not necessarily to those with weaker institutional support.

5. Result with Statistical Analysis

The statistical analysis is detailed and appropriately presented. The Necessary Condition Analysis (NCA) results and fsQCA sufficiency analysis are well-structured. The paper effectively presents configurations that lead to both high- and non-high-quality entrepreneurship. The authors also include robustness checks (e.g., sensitivity analysis), which enhances the credibility of the findings. However, the tables could be more reader-friendly with clearer labels and explanations of key terms.

6. Conclusion

The conclusion is comprehensive and ties the research findings back to the research questions. It offers clear policy recommendations, making it practical for a wide audience. However, the authors could emphasize future research directions more explicitly. For instance, what are the next steps in understanding the influence of institutional configurations on entrepreneurship?

7. Implication

The implications of the paper are well-stated. The study provides policymakers and stakeholders in emerging economies (e.g., China) with theoretical and practical insights into improving entrepreneurial quality among returnees. The institutional configurations that enhance entrepreneurship, such as the ‘opportunity-resource collaborative legitimacy drive,’ are highly relevant for governments and organizations seeking to boost economic development. The paper also highlights regional disparities and suggests tailored institutional frameworks, which is a strong practical contribution.

8. References

The reference section is well-organized and includes a wide range of sources. However, some recent influential works on entrepreneurship and institutional theory seem to be missing. Including more works from 2023 and 2024 would strengthen the paper's engagement.

9. English Grammar and Cohesiveness in Writing

The paper is generally well-written and cohesive, but there are a few areas where improvements in grammar and sentence structure could enhance readability. Here are some observations:

• Sentence Structure: Some sentences are lengthy and complex, making them harder to follow. Breaking them into shorter, clearer sentences would improve flow.

• Passive Voice: The paper uses the passive voice frequently. While this is common in academic writing, an occasional active voice would make the text more engaging.

• Cohesion: The transitions between sections are smooth, but adding more signposting within paragraphs could help guide the reader through the argument more clearly.

Example:

In the introduction, the transition between the discussion of institutional complexity and returnee entrepreneurship could be more seamless by using transitional phrases like, "Building on these insights, the current study explores...".

Summary of Improvements:

1. Incorporate 2023 and 2024 literature for contemporary relevance.

2. Simplify statistical tables and ensure they are more reader-friendly.

3. Shorten complex sentences and reduce over-reliance on the passive voice.

4. Expand the sample size to include more regions, improving the generalizability of findings.

5. Justify the selection of the 28 regions and discuss potential selection bias.

6. Consider using longitudinal data to capture changes over time in institutional environments.

7. Incorporate qualitative data through case studies or interviews with returnee entrepreneurs to provide richer insights.

8. Clarify how key constructs (e.g., entrepreneurial quality) are measured and acknowledge potential limitations.

Reviewer #2: How does the institutional environment improve the entrepreneurial quality of returnees?A configuration analysis based on a complex system view

I express profound gratitude to review the article.This study analyzes the complex relationship between institutional configurations and the quality of returnee entrepreneurship in China. The research samples include 28 regions. The subsequent issues pertain to the article are as follows.

Introduction

• Add research objectives and research questions in the introduction section in bullets.

Theoretical Background and Literature Review

• Add theoretical background. Add theories related to the topic.

• Add research Hypothesis in bullets,

• Before starting methodology there is a typing error” Fig 1. This is the Fig 1 Title.This si the Fig 1 Legend”.

•

Empirical Results Discussion and Implications

• Add the discussion part at the end of the empirical results. In which provide overall discussion regarding results and its impacts and implications.

6. PLOS authors have the option to publish the peer review history of their article (what does this mean? ). If published, this will include your full peer review and any attached files.

**Do you want your identity to be public for this peer review?** For information about this choice, including consent withdrawal, please see our Privacy Policy .

Reviewer #1: **Yes: ** Dr. Ahsan Riaz

Reviewer #2: **Yes: ** Dr Ameena Arshad

---

## [Author Response · Author response to Decision Letter 1]

13 Nov 2024

We are grateful for the editor and all reviewers for the constructive remarks and valuable suggestions, which have significantly improved the quality of our manuscript it. We have taken the reviewers’ suggestions into deep consideration and revised our work carefully regarding those.

Response to reviewer 1

Question 1�Incorporate 2023 and 2024 literature for contemporary relevance.

Response:Thank you very much for the reviewer's thorough evaluation! It is true that our paper lacked sufficient citation of the latest literature. In response to this suggestion, we have incorporated some relevant references from 2023-2024. We hope that our improvements will meet with your approval!

Question 2�Simplify statistical tables and ensure they are more reader-friendly.

Response:We are profoundly grateful for the reviewer's positive feedback on the content of the tables in our paper, especially for the recognition of the statistical analysis, Necessary Condition Analysis (NCA) results, fsQCA sufficiency analysis, and robustness checks. This feedback serves as a great encouragement for us.

We deeply appreciate the reviewer's suggestions concerning the table labels and explanations of key terms, and we attach great importance to them. Indeed, clear labels and explanations are crucial for readers to fully understand and grasp the content of the paper. While preparing the paper, We referred to the paradigms of existing QCA papers to set the table labels, aiming to maintain a high level of professionalism and standardization. However, We now realize that this approach may have inadvertently increased the reading difficulty for some readers.

To further enhance the readability and comprehensibility of the paper, we have provided explanatory examples below the tables, clarifying the meaning of the data presented. Please see the revised manuscript for details. We sincerely hope that our improvements will meet with your approval.

Question 3�Shorten complex sentences and reduce over-reliance on the passive voice.

Response: Thank you very much for your detailed review and constructive suggestions. We fully agree with the reviewer's observations and deeply appreciate their insights, which will undoubtedly assist us in further refining the readability and clarity of our paper.

Regarding sentence structure, we acknowledge that some sentences may have been overly long and complex, potentially hindering comprehension. We will take steps to break these down into shorter, more concise sentences to enhance the overall fluidity of the text.

Concerning the use of the passive voice, while it is prevalent in academic writing, we understand the reviewer's point that incorporating active voice occasionally can make the paper more engaging. We will review the text and make adjustments to balance the use of active and passive voices more effectively.

In terms of cohesion, while we have ensured smooth transitions between paragraphs, we recognize the value of adding more cues within paragraphs to further clarify our arguments. We will incorporate additional transitional phrases and clarifying statements to guide readers more seamlessly through our points.

To illustrate, in the introduction, we will use transitional phrases such as "Building on these insights, the current study explores..." to create a smoother transition between the discussion on institutional complexity and overseas returnee entrepreneurship.

Once again, we are deeply grateful for the reviewer's thoughtful and insightful feedback. We will carefully revise the paper according to these suggestions to enhance its readability and clarity. Thank you for helping us to further improve our work.

Question 4�Expand the sample size to include more regions, improving the generalizability of findings.

Question 5� Justify the selection of the 28 regions and discuss potential selection bias.

Response: Thank you very much for the reviewer's invaluable feedback, which is extremely precious. In response to the issues raised by the reviewer, we have carefully considered and provide the following reply.

Regarding the concerns of Limited Generalizability and Selection Bias, there are 31 provinces in mainland China (excluding Hong Kong, Macao, and Taiwan). We have selected 28 of them as our research subjects, which largely covers most regions of China. However, as you pointed out, there may indeed be some selection bias in our selection of 28 provinces. Therefore, we have made efforts to collect more data by altering the measurement methods of certain indicators. Our new sample size now covers 30 provinces, excluding Qinghai due to excessive missing data.

Furthermore, in terms of the Qualitative Comparative Analysis (QCA) method, QCA has a broad applicable sample size range, suitable for both small and large datasets. In our study, a sample of 30 provinces is fully applicable and can reveal complex relationships between institutional configurations and overseas returnees' entrepreneurship.

We understand every study has limitations. We have made our best efforts to ensure sample representativeness and research robustness. In future research, we aim to explore more diverse regions and cases to validate and expand our findings.We hope our enhancements meet your approval. Thank you!"

Question 6�Consider using longitudinal data to capture changes over time in institutional environments.

Response:Regarding the issue of the lack of longitudinal data, as you previously mentioned, utilizing cross-sectional data indeed poses challenges in demonstrating the evolutionary trends of how institutional environments influence the quality of entrepreneurship. In response to your suggestion, we have gathered relevant data spanning from 2013 to 2022 to conduct a multi-period Qualitative Comparative Analysis (QCA) within the framework of dynamic QCA. Based on the development trajectory of overseas returnees' entrepreneurship quality and policy support, we have divided this period into three distinct stages and conducted a thorough analysis to identify the institutional environment configurations that can lead to high-quality entrepreneurship among overseas returnees in each stage. Furthermore, we have rigorously examined the evolving trends in the importance of various institutional environment factors and the shifting configurations across these stages. To sum up, we mainly revised the Methodology, Results, Conclusions and policy implications of the paper. We sincerely hope that our enhancements will meet with your approval. Thank you!

Question 7�Incorporate qualitative data through case studies or interviews with returnee entrepreneurs to provide richer insights.

Response:Regarding your criticism concerning our 'over-reliance on quantitative methods,' we have conducted thorough reflections and considerations. Firstly, we wish to clarify that in utilizing research methodologies such as fsQCA and NCA, our intention was indeed to integrate qualitative and quantitative analyses. The QCA methodology itself constitutes a comprehensive framework that blends qualitative and quantitative elements, enabling us to extract complex causal relationships from data and reveal specific outcomes stemming from various combinations of conditions. Consequently, to a certain extent, our research has steered clear of an excessive reliance on quantitative methods, instead seeking a harmonious balance between quantitative analysis and qualitative understanding.

However, we fully comprehend and concur with your perspective that qualitative insights derived from interviews or case studies of overseas returnee entrepreneurs can offer nuanced factors and profound insights that statistical methods may overlook. These first-hand materials undeniably enrich our comprehension of the ecosystem surrounding overseas returnee entrepreneurship, shedding light on crucial aspects such as how overseas returnees experience and perceive institutional support during their entrepreneurial endeavors. Regrettably, due to resource and time constraints, we currently face difficulties in collecting sufficient first-hand interview data from overseas returnee entrepreneurs to directly incorporate qualitative case study content. For this, we offer our sincere apologies and acknowledge it as a limitation of this study. To address this shortfall, we have explicitly highlighted this limitation in the section devoted to research shortcomings and future prospects, and we are committed to overcoming this challenge in our future endeavors.

In the meantime, we have taken steps within our capabilities to mitigate the absence of qualitative analysis. Specifically, we have enriched the configuration analysis section by conducting a more in-depth qualitative examination of representative cases that align with the configuration characteristics. These cases not only provide further validation of our statistical findings but also unveil nuanced factors and intricate relationships that are elusive in quantitative analysis. We hope that our enhancements will be recognized by you. Thank you!

Question 8�Clarify how key constructs (e.g., entrepreneurial quality) are measured and acknowledge potential limitations.

Response:Regarding the operationalization of constructs, as you have pointed out, this study relies on indicators such as the number of companies listed on certain markets, which may not fully capture all aspects of enterprise quality (e.g., innovation, sustainability, or job creation). This feedback is indeed pertinent. The study integrates data from the Growth Enterprise Market (GEM) and Science and Technology Innovation Board (STAR Market) to measure the quality of overseas returnee entrepreneurship in the region, primarily based on the following three reasons:

First of all, the objectivity of market data is impotant. We choose the GEM and STAR markets as platforms to measure the quality of overseas returnee entrepreneurship. That is because these markets have strict listing standards and regulatory requirements. This way, they enable the screening of companies with high growth potential and innovative capabilities. The number of listed companies, as a measurement indicator, is objective and verifiable, directly reflecting the activity and market recognition of overseas returnee entrepreneurial activities in the region.

Moreover, the aim is to reflect the characteristics of innovation and high growth. The GEM and STAR markets particularly focus on the innovative capabilities and growth potential of startups. This aligns well with the typical characteristics of overseas returnee entrepreneurship (such as the application of new technologies and business model innovation). The number of companies listed on these markets can indirectly reflect the contributions of overseas returnee entrepreneurship in terms of innovation and high growth.

Furthermore, the basis is the data availability and representativeness. Listed company data is easily accessible and publicly transparent, facilitating cross-regional and cross-temporal comparisons and analyses. Companies in these markets often represent leaders and innovators in their industries, thus having high representativeness. In summary, although a single indicator (such as the number of listed companies) may not fully cover entrepreneurship quality (such as innovation, sustainability, job creation, etc.), it is one of the currently quantifiable and more direct indicators reflecting the quality and effectiveness of entrepreneurial activities. The listing standards of the GEM and STAR markets encompass considerations of multiple qualities such as corporate innovation capabilities, financial status, and market prospects. Therefore, this indicator can, to a certain extent, comprehensively reflect multiple dimensions of entrepreneurship quality.

In summary, although a single indicator (such as the number of listed companies) may not fully cover all aspects of entrepreneurial quality (such as innovation, sustainability, job creation, etc.), it is currently one of the quantifiable and relatively direct indicators reflecting the quality and effectiveness of entrepreneurial activities. The listing standards of the GEM and STAR Market inherently encompass considerations of multiple aspects such as enterprise innovation capabilities, financial conditions, and market prospects, so this indicator can, to a certain extent, comprehensively reflect multiple dimensions of entrepreneurial quality. Therefore, we believe that the current measurement method based on the number of companies listed on the GEM and STAR Market has certain advantages in scientificity and rationality. It relies on market data, objectively reflecting the contributions of overseas returnee entrepreneurship in terms of innovation and high growth, and possesses certain representativeness and comparability. Meanwhile, as you mentioned, this measurement method has some limitations, and it is crucial to continuously improve the measurement of overseas returnee entrepreneurial quality to make it more precise and comprehensive. Therefore, we have included the improvement of this indicator in our future outlook and look forward to continuously refining and optimizing this measurement system in future research.

In addition, we also add explicit descriptions of the measurement methods for some institutional elements.

Response to reviewer 2

Question 1�Add research objectives and research questions in the introduction section in bullets.

Response: Thank you very much for your meticulous review, esteemed reviewer! As you have pointed out, it is crucial to articulate the research objectives and research questions clearly in the introduction. We acknowledge that our expression in this section was not as definitive as it could have been. In response to your feedback, we have further refined the presentation of the research questions and research objectives in this part. We sincerely hope that our improvements will meet with your approval and enhance the clarity of the introduction. Thank you once again for your valuable and insightful feedback!

Question 2�Theoretical Background and Literature Review

• Add theoretical background. Add theories related to the topic.

• Add research Hypothesis in bullets.

Response: Thank you very much for your insightful suggestion, which we highly appreciate! As you mentioned, we did not sufficiently emphasize the institutional theory and transaction cost theory utilized in this paper within the theoretical background section. In response, we have enhanced our discussion on these theories in that part.

Furthermore, regarding your recommendation to include research hypotheses, we acknowledge its relevance, as some Qualitative Comparative Analysis (QCA) articles do present hypotheses according to their specific research needs. However, the primary objective of our study is to explore the institutional configurations that enable high-quality entrepreneurship among overseas returnees. In this context, some institutional factors have both positive and negative impacts on the quality of overseas returnee entrepreneurship. From a configurational perspective, we believe that these differences primarily arise from the varying roles these factors play when coupled with other institutional elements.

Therefore, in the theoretical background and literature review sections, we have focused on elucidating the impacts of different institutional factors on the quality of overseas returnee entrepreneurship, without formulating hypotheses. Instead, we provide a detailed analysis of the specific role each institutional factor plays in enhancing the quality of overseas returnee entrepreneurship, based on the configurational results presented in subsequent sections. We sincerely hope that our explanation gains your understanding. Thank you once again for your valuable feedback!

• Before starting methodology there is a typing error” Fig 1. This is the Fig 1 Title.This si the Fig 1 Legend”.

Response: I would like to express my deepest gratitude for your thoughtful reminder. The oversight was entirely on our part, and we have diligently made the necessary corrections in accordance with your instructions.

Question 3�Empirical Results Discussion and Implications

• Add the discussion part at the e

---

## [Decision Letter · Decision Letter 1]

PONE-D-24-40215R1How does the institutional environment improve the entrepreneurial quality of returnees? A configuration analysis based on a complex system viewPLOS ONE

Dear Dr. Yang,

Thank you for submitting your manuscript to PLOS ONE. After careful consideration, we feel that it has merit but does not fully meet PLOS ONE’s publication criteria as it currently stands. Therefore, we invite you to submit a revised version of the manuscript that addresses the points raised during the review process.

Please submit your revised manuscript by May 08 2025 11:59PM. If you will need more time than this to complete your revisions, please reply to this message or contact the journal office at plosone@plos.org . Please include the following items when submitting your revised manuscript:

We look forward to receiving your revised manuscript.

Kind regards,

Saranjam Baig

Academic Editor

PLOS ONE

Journal Requirements:

Reviewers' comments:

Reviewer's Responses to Questions

**Comments to the Author**

1. If the authors have adequately addressed your comments raised in a previous round of review and you feel that this manuscript is now acceptable for publication, you may indicate that here to bypass the “Comments to the Author” section, enter your conflict of interest statement in the “Confidential to Editor” section, and submit your "Accept" recommendation.

Reviewer #3: All comments have been addressed

Reviewer #4: All comments have been addressed

2. Is the manuscript technically sound, and do the data support the conclusions?

Reviewer #3: Yes

Reviewer #4: Yes

3. Has the statistical analysis been performed appropriately and rigorously? 

Reviewer #3: Yes

Reviewer #4: Yes

4. Have the authors made all data underlying the findings in their manuscript fully available?

Reviewer #3: Yes

Reviewer #4: Yes

5. Is the manuscript presented in an intelligible fashion and written in standard English?

Reviewer #3: Yes

Reviewer #4: Yes

6. Review Comments to the Author

Reviewer #3: (No Response)

Reviewer #4: reviews has given in the attached world file so it is enough to make required changes.

• Research Objectives/Questions needs to be highlighted properly. Research gap need to be addressed in the light of theory.

Review 2:

• Add research hypotheses and authors presenting model without any support of hypotheses.

• Connect theory with hypotheses. H1, H2, and H3 and discuss hypotheses in the light of the theory.

• Furthermore, provide hypotheses as you mentioned in Configuration Analysis for period-1 H1, (H1a), (H1b), and (H1c). Similarly in period 2 and 3.

Review 3:

• You normalize the data at first stage but how and which method you use to normalize the data. Please provide details. Also furnish detail about calibration method.

7. PLOS authors have the option to publish the peer review history of their article (what does this mean? ). If published, this will include your full peer review and any attached files.

**Do you want your identity to be public for this peer review?** For information about this choice, including consent withdrawal, please see our Privacy Policy .

Reviewer #3: No

Reviewer #4: No

---

## [Author Response · Author response to Decision Letter 2]

26 Mar 2025

Reviewer 2

Question1: Although the topic is very interesting but Research Objectives/Questions needs to be highlighted properly. Research gap need to be addressed in the light of theory.

Response: Thank you very much for your meticulous review, esteemed reviewer! As you have pointed out, it is crucial to highlight the research objectives/questions and theory gap clearly. We acknowledge that our expression in this section was not as definitive as it could have been. In response to your feedback, we have further refined the presentation of the research questions/objectives and theory gap. We sincerely hope that our improvements will meet with your approval and enhance the clarity of the introduction. Thank you once again for your valuable and insightful feedback!

Question2: (1)Add research hypotheses and authors presenting model without any support of hypotheses.

(2)Connect theory with hypotheses. H1, H2, and H3 and discuss hypotheses in the light of the theory.

(3)Furthermore, provide hypotheses as you mentioned in Configuration Analysis for period-1 H1, (H1a), (H1b), and (H1c). Similarly in period 2 and 3.

Response: Thank you for your in-depth review and valuable suggestions. We truly appreciate your emphasis on research rigor in suggesting to "add research hypotheses (H1, H2, H3, etc.) and connect theory with hypotheses." We fully understand the central role of hypothesis testing in traditional empirical research. However, based on the methodological characteristics of QCA (Qualitative Comparative Analysis) and our research design, we would like to explain in detail why QCA studies typically do not rely on traditional hypotheses.

Qualitative Comparative Analysis is a set-theory and Boolean-algebra-based method that explains complex social phenomena by identifying combinations of conditions ("configurations"). Its core idea is "causal complexity," where outcomes result from the joint action of multiple conditions rather than a single variable (Ragin, 2008). QCA emphasizes the following principles: ① Equifinality: The same outcome can be achieved through different combinations of conditions. ② Asymmetry: The combinations leading to the presence and absence of an outcome may differ. ③ Necessary and sufficient conditions: Analyze the necessity or sufficiency of conditions for an outcome using calibrated data (like fuzzy sets). QCA is suitable for medium-to-small-sample studies (N = 10 - 100) and is often used for exploratory (inductive) research to discover theoretical patterns through case comparison. It is not just a research method but a methodology. Based on the core ideas and principles of QCA methodology, research hypotheses are not required in papers applying this methodology for the following reasons:

First, the essential conflict between QCA's core value and hypothesis testing logic. Traditional hypothesis testing (e.g., regression analysis) assumes variables independently affect outcomes (e.g., H1: X→Y). In contrast, QCA identifies "causal recipes" (e.g., A + B + ∼C→Y), where conditions interact, substitute, or negate each other (Fiss, 2011). This combinational logic cannot be simplified into single hypotheses.

Second, causal complexity goes beyond linear assumptions. QCA deals with nonlinear relationships like "threshold effects" and "asymmetric paths." For example, "the combinations leading to high performance may differ from those leading to low performance" (Misangyi et al., 2017). Forcing linear hypotheses (e.g., A positively affects Y) may oversimplify phenomena and ignore dynamic dependencies between conditions.

Third, QCA is more for theory building than theory verification. It is often used in exploratory research to generate new theories through case comparison (e.g., discovering unrecognized success configurations). "QCA's counterfactual analysis supports theoretical innovation rather than verifying existing propositions" (Schneider & Wagemann, 2012). For instance, researchers might find that "firms with low resources achieve high growth through flexible strategies," a finding that cannot be derived from pre - specified hypotheses.

Fourth, methodological differences. QCA requires researchers to "return to cases" and interpret configurations' rationality using case knowledge (Ragin, 2008). For example, when analyzing multi - temporal data, researchers need to incorporate time dynamics (e.g., trajectory analysis as proposed by Pagliarin & Gerrits, 2020) rather than rely on assumption - driven statistical tests. Including hypothesis testing in the literature review might lead to "method-theory disconnection" (Park et al., 2020).

Moreover, most QCA papers in top journals do not pre-specify research hypotheses, such as Leppänen et al. (2021) in Academy of Management Journal and Howell et al. (2022) in Organization Science.

In summary, considering QCA methodology's characteristics and the prevalent writing patterns in top-journal QCA papers, we have decided not to add a research hypothesis section. We hope our explanation is understandable to you. Thank you!

Question3: You normalize the data at first stage but how and which method you use to normalize the data. Please provide details. Also furnish detail about calibration method.

Response: Thank you very much for the reviewer's thorough evaluation! We fully appreciate the importance of clearly describing data normalization and calibration methods. In response to your suggestions, we will elaborate on our data preprocessing steps and rationale. We've also enhanced the calibration section, specifying the software, function, and method used in the calibration process. We hope that our enhancements will meet with your approval. Thank you!

Editor

Question1: In-text citations problems persist like name of author and brackets with number

Fiss (2011)[66]

Response: Thank you for the in-depth review. We apologize for the oversight in the manuscript. We've carefully reread and revised the incorrect literature citations. You can see the specific revisions in the modified manuscript. We hope our revisions meet your approval. Thank you again!

Question2: Formatting problems persist in headings and proper numbering is missing

Response: Thank you for your reminder. We have carefully rechecked the numbering of headings in the manuscript and made appropriate adjustments where necessary. You can see the specific revisions in the modified manuscript. We hope our revisions meet your approval. Thank you again!

Question3: Important research papers that considered to be base papers are missing e.g.

Returnee entrepreneurs and the institutional environment: case study insights from ChinaJanuary 2019 International Journal of Emerging Markets 14(1):207-230 DOI:10.1108/IJoEM-11-2017-0504

Bomzon SD, Mukherjee S, Murti AB. The Process of Returnee Entrepreneurship: From Firm Creation to Contribution. InAcademy of Management Proceedings 2024 (Vol. 2024, No. 1, p. 18083). Valhalla, NY 10595: Academy of Management.

Response: Thank you for your important suggestion regarding the literature citations. As you pointed out, we had overlooked two key studies on returnee entrepreneurship. We have now added citations for these two papers in the manuscript. You can see the specific revisions in the modified manuscript. Thank you very much!

Question4: Repetition of concepts and words in whole article. Please give a proper read and omit them.

Response: Thank you very much for your careful review. We acknowledge the redundancy issues in the manuscript. We've carefully proofread the full text and addressed the redundancies by replacing repeated terms, standardizing abbreviations, and simplifying sentences. You can see the specific revisions in the modified manuscript. We hope these improvements meet your approval. Thanks again!

---

## [Editor Report · Decision Letter 2]

How does the institutional environment improve the entrepreneurial quality of returnees?

A configuration analysis based on a complex system view

PONE-D-24-40215R2

Dear Dr. Yang,

We’re pleased to inform you that your manuscript has been judged scientifically suitable for publication and will be formally accepted for publication once it meets all outstanding technical requirements.

Kind regards,

Saranjam Baig

Academic Editor

PLOS ONE

---

## [Editor Report · Acceptance letter]

PONE-D-24-40215R2

PLOS ONE

Dear Dr. Yang,

I'm pleased to inform you that your manuscript has been deemed suitable for publication in PLOS ONE. Congratulations! Your manuscript is now being handed over to our production team.

Kind regards,

on behalf of

Dr. Saranjam Baig

Academic Editor

PLOS ONE